# Application of Metagenomic Next-Generation Sequencing in the Diagnosis of Pneumonia Caused by *Chlamydia psittaci*

Junli Tang,[a] Wanmei Tan,[a] Lingxin Luo,[a] Huan Xu,[b] Na Li[a]

aDepartment of Respiratory and Critical Care Medicine, The Second Affiliated Hospital of Chongqing Medical University, Chongqing, People's Republic of China
bVision Medicals Center for Infection Disease, Guangzhou, People's Republic of China

**ABSTRACT** Psittacosis is an uncommon disease which mainly presents as community-acquired pneumonia (CAP). We aim to apply metagenomic next-generation sequencing (mNGS) as a promising tool in the diagnosis of psittacosis pneumonia and to describe its clinical spectrum to provide physicians with a better understanding and recognition of this disease. Thirteen cases of psittacosis pneumonia were diagnosed by using mNGS. A retrospective analysis of the data on clinical manifestations, laboratory data, computed tomography (CT) images, new diagnosis tools, treatments, and outcomes was summarized. These patients had common symptoms of fever and weakness; some had poor appetite, cough, myalgia, and headache. Ten patients developed acute respiratory distress syndrome (ARDS), among which six patients were severe pneumonia cases and needed ventilator therapy. Most patients got psittacosis pneumonia during the cold season. Ten cases were sporadic, but three were family clustering. All of the 13 patients were traced to an exposure history to birds, cat, or poultry, among which 2 only touched the innards of killed poultry before cooking, which may be an atypical exposure history not been reported before, to our knowledge. Most patients had various degrees of liver dysfunction. Air-space consolidations, along with ground-glass opacities and reticular shadows, were detected on chest CT scan. mNGS takes 48 to 72 h to provide results and helps to diagnose psittacosis. After being diagnosed by mNGS, with effective medicines, all patients finally had complete recoveries. The use of mNGS can improve the diagnostic rate of psittacosis pneumonia and shorten the course of disease control.

**IMPORTANCE** Psittacosis pneumonia is easily underdiagnosed and misdiagnosed. In this study, we use mNGS in the diagnosis of psittacosis pneumonia. We found this disease is prone in the cold season, and touching the innards of killed poultry during cooking may be an atypical exposure history which has not been reported before to our knowledge. There are sporadic cases and family outbreak cases as well. Except for typical symptoms of fever and weakness, headache may be the main and only symptom in some patients. The rate of severe pneumonia is high among inpatients with psittacosis pneumonia, and the incidence of hepatic involvements is also high. Psittacosis pneumonia can be cured if the diagnosis is accurate and in time, even if it is severe pneumonia on admission. Some problems worthy of our attention about psittacosis pneumonia were put forward, such as its sick season, special exposure history, the rate of severe disease, and the high cure rate. mNGS can quickly and objectively detect more rare pathogenic microorganisms in clinical specimens without the need for specific amplification and has an advantage in the diagnosis of rare pathogenic bacteria in difficult cases such as psittacosis pneumonia. The use of mNGS can improve the accuracy and reduce the delay in the diagnosis of psittacosis, which shortens the course of disease control.

**KEYWORDS** psittacosis, *Chlamydia psittaci*, metagenomic next-generation sequencing (mNGS), diagnosis, pneumonia

Address correspondence to Na Li, 300281@hospital.cqmu.edu.cn.

The authors declare no conflict of interest.

Psittacosis (ornithosis) is a naturally occurring infectious disease caused by *Chlamydia psittaci* in birds, poultry, and some mammals (1). *C. psittaci* is traditionally regarded as an avian pathogen of global distribution (2). Pathogens can be transmitted to human beings through inhalation of aerosols from contaminated bird substances such as droppings, plumage, or tissue (3). Psittacosis in humans usually manifests as pneumonia, which ranges in severity from asymptomatic to fatal (4), and few clinically diagnosed cases have been reported because of the lack of accurate diagnostic methods.

The flulike atypical pneumonia symptoms and exposure history to birds are the primary criteria for clinical diagnosis. Laboratory diagnosis requires meeting any one of the following three criteria: (i) isolation of *Chlamydia psittaci* from respiratory secretions, (ii) a 4-fold or greater increase in antibody titer between serum samples collected 2 weeks apart, using a complement fixation test (CFT) or microimmunofluorescence (MIF), and (iii) IgM antibody against *C. psittaci* titer detected by MIF of 1:16 or higher (5). Psittacosis can be diagnosed by culture, but *C. psittaci* grows slowly in culture. Furthermore, biosafety level 3 facilities and cell cultures are required, which are not available in most medical microbiology laboratories (6). Because of its nonspecific symptoms and the limitations of current tests, psittacosis is easily underdiagnosed and misdiagnosed (7).

Metagenomic next-generation sequencing (mNGS) is a culture-independent technology that has been widely utilized in microbiology research but not in routine clinical microbiological diagnostics. The application of mNGS technology and its various methodological variants now makes it possible to detect different types of microorganisms present within a microbial sample simultaneously, using a culture-independent approach and in a single sequencing run (8). Recently, there have been more applications of mNGS in clinical diagnosis. It can help the diagnosis of pathogenic microbe infections in various systems, such as nervous system, gastrointestinal system, respiratory system (including psittacosis pneumonia), etc. (9, 10).

In this article, we analyzed a cohort of 13 inpatients in which mNGS for psittacosis diagnosis was incorporated into the diagnostic algorithm. We describe the clinical features, laboratory data, computed tomography, treatments, response to treatment, and outcomes of psittacosis pneumonia. Furthermore, we evaluate the contribution of mNGS as an effective method for establishing the diagnosis.

## RESULTS

**Patients' characteristics.** Five women and eight men with *C. psittaci* pneumonia were identified. Their median age was 61.8 years (range, 46 to 74). Six of them had underlying diseases such as diabetes, hypertension, or hepatitis B. Two of them had a smoking history (Table 1).

**Exposure history.** All of the 13 patients (100%) had a history of close contact with birds, poultry, or cats. They had raised birds, poultry, or cat or had a contact history such as visits to live poultry markets and touching the innards of killed poultry during cooking (Table 2). It is worth noting that three patients presented with family cluster onsets, all of whom were in charge of taking care of two pet parrots, which died 1 week before the patients' admissions.

**Symptoms.** All patients had similar symptoms, including recurrent fever (100%) higher than 38.5°C and weakness (100%). Twelve patients had poor appetite (92.3%), six patients had cough (46.2%), five patients had myalgia (38.5%), and three patients had headache (23.08%) over the course of the disease. It is worth noting that one of the patients was admitted to the department of neurology with headache as the main symptom and was later referred to the department of respiratory medicine for pneumonia. There were three patients with P/F ($PaO_2/FiO_2$ ratio) higher than 300 (23.08%), seven patients with P/F greater than 200 and less than 300 (53.84%), and three patients with P/F greater than 100 and less than 200 (23.08%). Six patients were severe pneumonia cases (46.2%), and ten patients developed acute respiratory distress syndrome (ARDS) (76.92%). Six patients needed ventilator therapy (46.2%) over the course of the disease, and two of them were given invasive ventilator therapy (15.4%) (Table 3).

**TABLE 1** Patient characteristics[a]

| Patient no. | Male or female | Age (median [range] [yrs]) | Underlying disease(s) | Smoker |
|---|---|---|---|---|
| 1 | M | 65 | Hypertension | Yes |
| 2 | F | 64 | Hypertension | No |
| 3 | M | 74 | Hepatitis B | No |
| 4 | M | 73 | Diabetes, hypertension | Yes |
| 5 | M | 70 | Diabetes, hypertension | No |
| 6 | M | 49 | None | No |
| 7 | F | 68 | None | No |
| 8 | M | 58 | None | No |
| 9 | F | 50 | None | No |
| 10 | F | 72 | None | No |
| 11 | M | 46 | Hepatitis B | No |
| 12 | F | 48 | None | No |
| 13 | M | 72 | None | No |
| Total | 8 M, 5 F | 61.8 (46–74) | 6/13 | 2/13 |

[a]The data in the "total" row indicate the proportions of patients unless otherwise indicated. M, male; F, female.

**Dates of admission and durations of disease and time to discharge.** The dates of admissions were from June 2020 to May 2021. Ten patients fell ill in cold season (76.9%). The longest duration from the onset of the illness to admission was 30 days, and the longest duration from admission to discharge was 34 days; the shortest duration from admission to discharge was 5 days. All 13 patients were diagnosed by mNGS, and all of them experienced complete recoveries. The earlier the mNGS test was done, the shorter the course of the disease (Table 4).

**mNGS results.** Blood and sputum samples or bronchoalveolar lavage fluid (BALF) were collected for mNGS, and the results of mNGS are shown in Table 5. Not all patients were tested with both blood and respiratory samples. For diagnosis, BALF collection can better exclude the possibility of sample contamination because it extracts sputum from the bronchi and reflects the real pathogenic bacterial infection of the lungs. However, bronchoalveolar lavage requires bronchoscopy; some patients were too old or too seriously ill to tolerate bronchoscopy, and some patients were unwilling to complete bronchoscopy for their own reasons. Therefore, for these patients, we chose their sputum or blood as testing specimens.

*C. psittaci* is an obligate intracellular bacterium which must grow and reproduce in living cells both *in vivo* and *in vitro*. Therefore, the detection sensitivity and detection rate of intracellular bacteria are relatively low due to the small number of bacteria released extracellularly into body fluids such as blood, sputum, and bronchoalveolar lavage fluid. Because of the difficulty of DNA extraction and low possibility of contamination, the *C. psittaci* was considered detected if (i) its genus was among the top 20 with the highest standardized specifically mapped read number (SDSMRN), (ii) it ranked first

**TABLE 2** Exposure history

| Patient no. | Exposure history |
|---|---|
| 1 | Raised pigeons |
| 2 | Raised parrot |
| 3 | Raised ducks |
| 4 | Raised pigeons |
| 5 | Contact with duck, touched the innards of killed poultry during cooking |
| 6 | Raised chickens |
| 7 | Contact with chicken, touched the innards of killed poultry during cooking |
| 8 | Raised cat |
| 9 | Contact with parrot |
| 10 | Raised parrot |
| 11 | Contact with parrot |
| 12 | Raised ducks |
| 13 | Contact with chicken during visits to live poultry markets |

**TABLE 3** Symptoms of psittacosis pneumonia

| Patient no. | Fever | Weakness | Poor appetite | Cough | Chill | Myalgia | Headache | Critically ill | ARDS | P/F | Use of ventilator |
|---|---|---|---|---|---|---|---|---|---|---|---|
| 1 | Yes | Yes | Yes | Yes | No | No | No | Yes | Yes | 222.5 | Yes |
| 2 | Yes | Yes | Yes | Yes | No | No | Yes | No | Yes | 189 | No |
| 3 | Yes | Yes | Yes | No | No | Yes | No | Yes | Yes | 215 | Yes |
| 4 | Yes | Yes | Yes | No | No | No | No | Yes | Yes | 236 | Yes (invasive) |
| 5 | Yes | Yes | Yes | No | No | No | No | No | Yes | 224 | No |
| 6 | Yes | Yes | Yes | Yes | No | No | No | Yes | Yes | 134 | Yes (invasive) |
| 7 | Yes | Yes | Yes | No | No | Yes | No | Yes | Yes | 110 | Yes |
| 8 | Yes | Yes | Yes | Yes | No | Yes | Yes | No | No | 452 | No |
| 9 | Yes | Yes | No | Yes | No | Yes | No | No | No | 457 | No |
| 10 | Yes | Yes | Yes | No | Yes | No | No | No | Yes | 206 | No |
| 11 | Yes | Yes | Yes | No | No | No | No | No | Yes | 281 | No |
| 12 | Yes | Yes | Yes | Yes | No | Yes | Yes | Yes | Yes | 224 | Yes |
| 13 | Yes | Yes | Yes | No | Yes | No | No | No | No | 361 | No |
| Total (no. of positive patients/total no. of patients [%]) | 13/13 (100) | 13/13 (100) | 12/13 (92) | 6/13 (46) | 2/13 (15) | 5/13 (38) | 3/13 (23) | 6/13 (46) | 10/13 (77) | | 6/13 (46) |

within its genus, and (iii) it had an SDSMRN of >1 (11). Therefore, in the interpretation of mNGS results, even though the DNA sequence amounts of intracellular bacteria were small, We should also think of it as a pathogen that is always highly likely to cause disease.

*C. psittaci* was detected by mNGS in all patients. mNGS can detect not only *C. psittaci* but also other microorganisms at the same time. Some are pathogenic bacteria such as *Haemophilus parainfluenzae*, and some are colonizing bacteria such as *Lactobacillus salivarius*, *Enterococcus faecalis*, etc. It was found that fewer colonizing bacteria were detected in BALF and blood, while a large number of colonizing bacteria were detected in sputum, such as in case 3 and case 10. This is because sputum is discharged through the mouth, and sputum specimens are susceptible to contamination by oropharyngeal colonizing bacteria.

We listed the coverage of mNGS in some patients. As we found, 3,188 species-specific *C. psittaci* sequences covered by 8.82% of the *C. psittaci* genome were detected by mNGS in the sputum sample of the third patient. Seventeen specific *C. psittaci* sequences that covered 0.09% of the total *C. psittaci* genome were detected by mNGS in the blood sample of the sixth patient, and 269 specific *C. psittaci* sequences that covered 1.51% of the total *C. psittaci* genome were detected by mNGS in the BALF sample of the sixth patient. A total of 963 specific *C. psittaci* sequences that covered 5.13% of the total *C. psittaci* genome were detected by mNGS in the BALF sample of the seventh patient. Three hundred seventy-nine specific *C. psittaci* sequences that covered 2.69% of the total *C. psittaci* genome

**TABLE 4** Date of admission and durations of disease progression

| Patient no. | Date of admission (yr-mo-day) | No. of days from onset of illness until hospital admission | No. of days from hospital admission until departure | Date of mNGS test (yr-mo-day) | No. of days from onset of illness until mNGS test | No. of days from onset of illness until hospital departure |
|---|---|---|---|---|---|---|
| 1 | 2020-06-11 | 30 | 13 | 2020-06-15 | 34 | 43 |
| 2 | 2020-06-27 | 3 | 24 | 2020-07-08 | 14 | 27 |
| 3 | 2020-10-16 | 4 | 21 | 2020-10-20 | 8 | 25 |
| 4 | 2020-11-01 | 10 | 12 | 2020-11-01 | 10 | 22 |
| 5 | 2020-12-10 | 3 | 12 | 2020-12-15 | 8 | 15 |
| 6 | 2020-12-19 | 2 | 34 | 2020-12-24 | 7 | 36 |
| 7 | 2021-01-10 | 12 | 26 | 2021-01-11 | 13 | 38 |
| 8 | 2021-01-24 | 5 | 28 | 2021-01-26 | 7 | 33 |
| 9 | 2020-12-23 | 7 | 5 | 2020-12-25 | 9 | 12 |
| 10 | 2020-12-22 | 4 | 9 | 2020-12-24 | 6 | 13 |
| 11 | 2020-12-28 | 3 | 10 | 2020-12-29 | 4 | 13 |
| 12 | 2020-10-30 | 7 | 20 | 2020-11-02 | 9 | 27 |
| 13 | 2021-05-13 | 7 | 7 | 2021-05-14 | 8 | 14 |

**TABLE 5** mNGS results

| Patient | Platform | Sample type(s) | Data vol (no. of reads) | Detected pathogen(s) (no. of species-specific reads) |
|---|---|---|---|---|
| 1 | NextSeq 550Dx sequencing platform | BALF[a] | 13,589,283 | *Chlamydia psittaci* (3), human gammaherpesvirus 4 (33) |
| 2 | NextSeq 550Dx sequencing platform | BALF | 12,282,056 | *Chlamydia psittaci* (2) |
| 3 | NextSeq 550Dx sequencing platform | Sputum | 10,795,185 | *Chlamydia psittaci* (3,188), *Enterococcus faecalis* (180), *Tropheryma whipplei* (4), *Candida glabrata* (14,648), human papillomavirus 100 (1,529), torquetenovirus (6), human gammaherpesvirus 4 (14) |
| 4 | NextSeq 550Dx sequencing platform | BALF | 15,837,687 | *Chlamydia psittaci* (68), *Enterococcus faecium* (12), *Candida albicans* (524), human alphaherpesvirus 1 (4) |
| 5 | NextSeq 550Dx sequencing platform | BALF | 19,072,595 | *Chlamydia psittaci* (7), *Haemophilus parainfluenzae* (143), human betaherpesvirus 7 (23) |
| 6 | NextSeq 550Dx sequencing platform | BALF | 19,466,389 | *Chlamydia psittaci* (269) |
| | | Blood | 18,357,921 | *Chlamydia psittaci* (17), human gammaherpesvirus 4 (18) |
| 7 | NextSeq 550Dx sequencing platform | BALF | 19,528,945 | *Chlamydia psittaci* (963), *Haemophilus influenzae* (23), *Pasteurella multocida* (30), *Candida albicans* (12) |
| 8 | NextSeq 550Dx sequencing platform | BALF | 21,669,259 | *Chlamydia psittaci* (107) |
| 9 | NextSeq 550Dx sequencing platform | BALF | 9,406,008 | *Chlamydia psittaci* (2), *Haemophilus parainfluenzae* (97), human betaherpesvirus 7 (3) |
| 10 | NextSeq 550Dx sequencing platform | Blood | 14,781,661 | *Chlamydia psittaci* (2) |
| | | Sputum | 10,067,954 | *Chlamydia psittaci* (379), *Lactobacillus gasseri* (10,941), *Lactobacillus salivarius* (4,493), *Enterococcus faecalis* (1,296), *Tropheryma whipplei* (191), *Haemophilus parainfluenzae* (203), *Candida albicans* (2,101) |
| 11 | NextSeq 550Dx sequencing platform | BALF | 14,159,894 | *Chlamydia psittaci* (563) |
| 12 | BGISeq-50/MGISeq-2000 platform | BALF | 62,745,758, | *Chlamydia psittaci* (1,337), human alphaherpesvirus 1 (4) |
| 13 | NextSeq 550Dx sequencing platform | BALF | 60,736,265 | *Chlamydia psittaci* (108), *Candida albicans* (74), human betaherpesvirus 5 (4) |

[a]BALF, bronchoalveolar lavage fluid.

were detected by mNGS in the sputum sample of the tenth patient. A total of 563 specific *C. psittaci* sequences that covered 2.24% of the total *C. psittaci* genome were detected by mNGS in the BALF sample of the 11th patient (Table 5 and Fig. 1).

**Orthogonal testing.** We tried to use the remaining extracted DNA nucleic acid from three patients' samples to do PCR testing and Sanger sequencing for *C. psittaci* in the laboratory (12). The results of PCR and Sanger sequencing verified the existence of *C. Psittaci* and were consistent with the results of mNGS.

In patient 11 in whose BALF 779 specific *C. psittaci* sequences and no other pathogen were detected by mNGS, the PCR amplification bands (no. 20884) were clear (Fig. 2), the Sanger sequencing results had a query coverage of 95% and a percent identity of 100%, and the E value was 0.0 by BLAST comparison with *C. psittaci* (supplemental file 1).

In patient 10 in whose sputum 379 specific *C. psittaci* sequences and another oral colonizing pathogen were detected by mNGS, the PCR amplification bands (no. 20638) were clear (Fig. 2), but the Sanger sequencing results could not be verified by BLAST due to the mixture with other pathogen sequences. However, in the blood sample of patient 10 in which only 3 specific *C. psittaci* sequences and no other pathogens were detected by mNGS, the PCR amplification bands (no. 7262) were not found (Fig. 2).

In patient 9 in whose BALF only 2 specific *C. psittaci* sequences were detected by mNGS, the PCR amplification bands (no. 20707) were not found (Fig. 2) because the number of *C. psittaci* genomic DNA sequences was too small to be amplified by PCR.

Because it is a retrospective study, not all the patients had enough remaining samples for PCR testing and Sanger sequencing for *C. psittaci* to be done in the laboratory. But the above-described results of PCR and Sanger sequencing verified the existence of *C. psittaci* and are consistent with the results of mNGS, which can have certain representativeness.

**Laboratory examinations.** The indicators of infection increased as follows: 4 patients had increased leucocytes (30.77%), 12 patients (92.3%) had increased neutrophils with a decreased percentage of lymphocytes, and all patients had increased C-reactive protein (CRP) and procalcitonin (PCT).

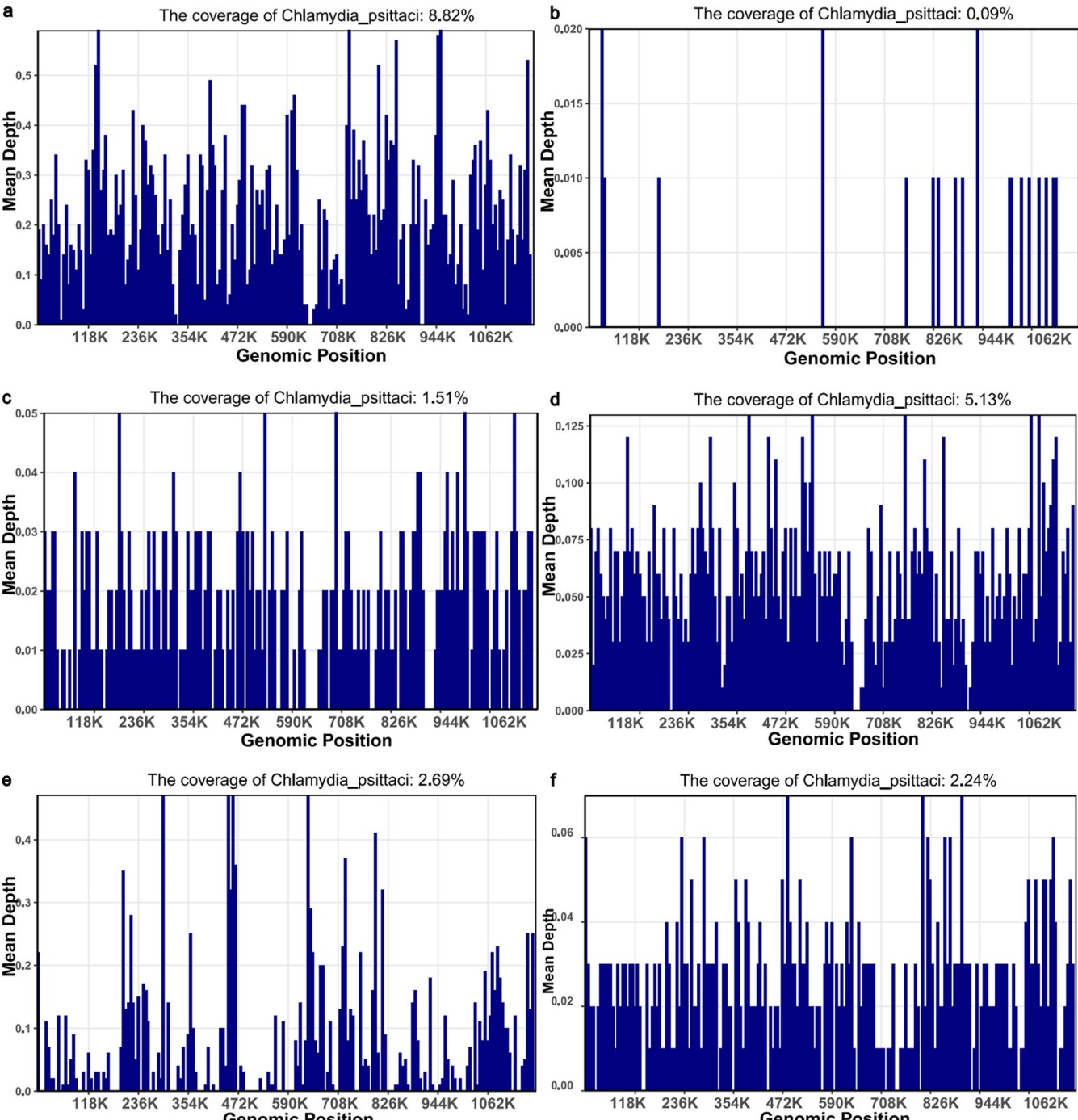

**FIG 1** Metagenomic next-generation sequencing results of some patients. (a) Sputum sample of patient 3; (b) blood sample of patient 6; (c) BALF sample of patient 6; (d) BALF sample of patient 7; (e) sputum sample of patient 10; (f) BALF sample of patient 11.

Some patients had hepatic dysfunction, including 8 patients (61.5%) with increased γ-glutamyl transpeptidase (GGT), 11 patients (84.6%) with increased alanine aminotransferase (ALT), and 12 patients (92.3%) with increased aspartate aminotransferase (AST). Seven patients (53.8%) had an increase in creatine kinase (CK).

All patients increased in plasma D-dimer levels. Eight patients (61.5%) had hypokalemia. Six patients were tested for (1-3)-β-d-glucan, and this index was elevated in one of them.

After treatment, patients' infection indexes decreased or even returned to normal, and plasma D-dimer levels decreased significantly. Some patients were not tested for

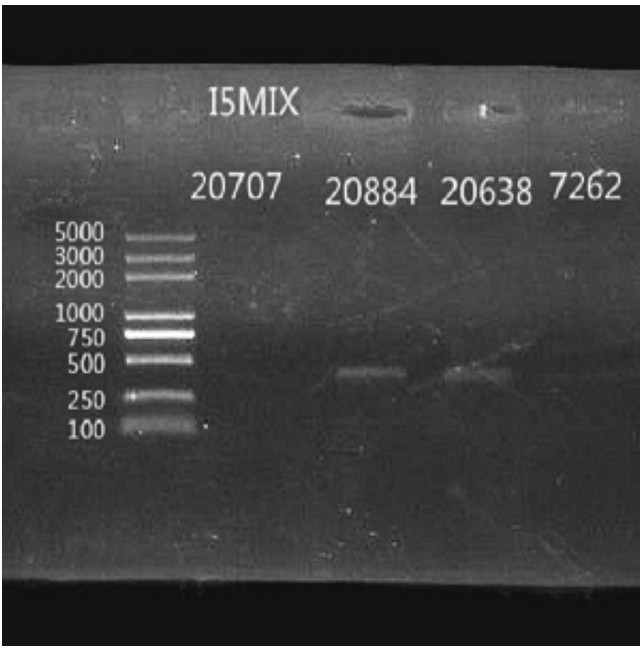

**FIG 2** PCR amplification results of 3 patients' samples. No. 20707 is DNA nucleic acid extracted from the BALF of patient 9 with 2 specific *C. psittaci* sequences found by mNGS. No. 20884 is DNA nucleic acid extracted from the BALF of patient 11 with 779 specific *C. psittaci* sequences found by mNGS. No. 20638 is DNA nucleic acid extracted from the sputum of patient 10 with 379 specific *C. psittaci* sequences found by mNGS. No. 7262 is DNA nucleic acid extracted from the blood of patient 10 with 3 specific *C. psittaci* sequences found by mNGS. The target size of PCR product is 393 bp.

creatine kinase after treatment, but those who were tested had lower creatine kinase levels than before. Their liver function was better than before treatment, and their low potassium levels were corrected (Table 6).

**Radiological examination.** In their chest computed tomography (CT) examinations, all patients had inflammatory lesions, but not all of their inflammatory lesions can be observed in the same lobe of lung. Eight patients' lesions began in the lower lobe of lung, but the others' lesions did not. Six of them could be observed as consolidation with air bronchograms. Infiltrates, reticular shadows, and consolidations with bronchograms can be seen in their chest CT examinations (Fig. 3).

**Treatment.** Prior to the discovery of *Chlamydia psittaci* pathogen infection, the patients were treated with empirical antibiotics in accordance with community-acquired pneumonia guidelines, but their clinical condition did not improve or even gradually deteriorated. After their infection with *Chlamydia psittaci* was confirmed by mNGS and the corresponding drugs were added, their condition began to improve, and they were discharged successfully. Not all patients had only psittacosis pneumonia disease. Through mNGS testing, we found that some patients were also infected by other pathogens. For these patients, we also had treatments that targeted other pathogens, so their treatment was not the same. For example, for viral infections, we added antiviral therapy; for fungal infections, we added antifungal therapy; and for bacterial infection, we applied antibiotics accordingly (Table 7). All patients received treatment for psittacosis pneumonia; they were treated with antibiotic therapy with moxifloxacin or tetracyclines according to the community-acquired pneumonia management guidelines (4, 13). Nine of them were treated with quinolones, among which only four responded, and the remaining irresponsive five patients were switched to tetracyclines and then cured. It is worth mentioning that because some patients were infected with other pathogens and the presence of other pathogens was discovered at the beginning, the possibility of *Chlamydia psittaci* infection was ignored, so only drugs targeted at other pathogens were used, but the patients' condition did not improve. It was only when drugs for psittacosis were added that their condition slowly improved, and finally, all patients recovered (Table 7).

**TABLE 6** Laboratory examination results[a]

| Patient no. | WBC (3.5–9.5 g/L) | | N% (45–75%) | | L% (20–50%) | | CRP (<10 mg/L) | | PCT (0.02–0.0505 ng/mL) | | Plasma D-dimer (0–550 ng/mL) | | CK (38–174 U/L) | | GGT (10–60 U/L) | | ALT (7–40 U/L) | | AST (13–35 U/L) | | Hypokalemia (3.5–5.2 mmol/L) | | (1-3)-β-D-Glucan BTRT (<37.5) | (<70 pg/mL) ATRT |
|---|---|---|---|---|---|---|---|---|---|---|---|---|---|---|---|---|---|---|---|---|---|---|---|---|
| | BTRT | ATRT | BTRT | ATRT | BTRT | ATRT | BTRT | ATRT | BTRT | ATRT | BTRT | ATRT | BTRT | ATRT | BTRT | ATRT | BTRT | ATRT | BTRT | ATRT | BTRT | ATRT | | |
| 1 | 10.7 | 4.65 | 90.8 | 49.9 | 6.7 | 38.3 | 93.05 | <5.0 | 0.084 | 0.02 | 1,410.6 | | 55 | | 217 | 145 | 50 | 48 | 46 | 38 | 3.3 | 3.72 | <37.5 | <37.5 |
| 2 | 8.13 | 5.01 | 84.3 | 57.9 | 9.6 | 26.3 | 112.62 | 5.43 | 0.283 | 0.051 | 3,235.1 | 2115 | 72 | | 89 | 109 | 44 | 30 | 42 | 29 | 2.96 | 4.37 | <37.5 | <37.5 |
| 3 | 10.1 | 6.33 | 92.3 | 89.8 | 3.3 | 5.4 | 141 | 10.96 | 1.16 | 0.06 | 685.1 | | 3,279.8 | | 11 | 45 | 33 | 46 | 113 | 31 | 3.32 | 4.08 | 109.8 | |
| 4 | 7.85 | 6.01 | 84 | 76 | 8.9 | 14.5 | 149.75 | 19.55 | 0.162 | 0.223 | 8,263.9 | 2,678.9 | 300 | 624 | 244 | 138 | 107 | 54 | 131 | 42 | 4.22 | 3.99 | | |
| 5 | 10.2 | 5.31 | 89.8 | 61.8 | 6.2 | 28.9 | >200 | 5.48 | 1.07 | | 1,666.8 | | 332 | | 57 | 77 | 130 | 97 | 228 | 119 | 3.93 | 3.84 | | |
| 6 | 4.59 | 5.44 | 95.5 | 59.3 | 3 | 27.5 | 176.55 | <0.5 | 16.64 | 0.04 | 3,318.9 | 365.6 | 4,434 | 278.9 | 78 | 50 | 185 | 16 | 624 | 15 | 3.24 | 4.31 | <37.5 | |
| 7 | 7.33 | 8.35 | 95.5 | 65.8 | 3 | 20.7 | >200 | <5.0 | 0.53 | 0.05 | 2,480.1 | 1,061.1 | 15 | | 26 | 38 | 132 | 23 | 106 | 20 | 3.33 | 4.27 | <37.5 | |
| 8 | 6.84 | 5.72 | 85.9 | 63.6 | 8.1 | 26.9 | 133.03 | <5.0 | 0.13 | 0.03 | 213.2 | | 284 | | 156 | 52 | 102 | 36 | 47 | 18 | 3.55 | 3.74 | <37.5 | |
| 9 | 4.3 | 5.14 | 58.8 | 66.7 | 31.4 | 24.9 | 56.81 | 10.82 | 0.089 | 0.045 | 990.6 | 593.5 | 49 | 38 | 173 | | 267 | | 108 | | 4.32 | | | |
| 10 | 9.92 | 6.9 | 86.7 | 64.7 | 6.8 | 26.4 | >200 | 27.75 | 0.641 | 0.053 | 1,294.6 | 829.3 | 755 | 54 | 41 | 165 | 54 | 87 | 91 | 88 | 2.81 | 4.6 | | |
| 11 | 7.83 | 4.47 | 83.5 | 64.9 | 11.5 | 25.3 | 155.13 | 6.87 | 0.64 | 0.024 | 1,059.1 | 517.8 | 100 | | 14 | | 23 | | 31 | | 4.29 | 4.08 | | |
| 12 | 5.64 | 6.79 | 91.8 | 77.7 | 6.4 | 15.8 | >200 | 27.92 | 0.993 | 0.075 | 689.3 | 516 | 98 | | 101 | 68 | 113 | 64 | 143 | 55 | 3.65 | 4.64 | | |
| 13 | 5.57 | 5.86 | 84.4 | 70.8 | 9.3 | 17.6 | >200 | 73.46 | 0.84 | 2.66 | 840.6 | | 349.5 | | 38 | 39 | 69 | 69 | 118 | 68 | 3.26 | 4.28 | | |

[a]WBC, white blood cell; N%, Percentage of neutrophils; L%, Percentage of lymphocytes; CRP, C-reactive protein; PCT, procalcitonin; CK, creatine kinase; GGT, γ-glutamyl transpeptidase; ALT, alanine aminotransferase; AST, aspartate aminotransferase; BTRT, before treatment; ATRT, after treatment (normal).

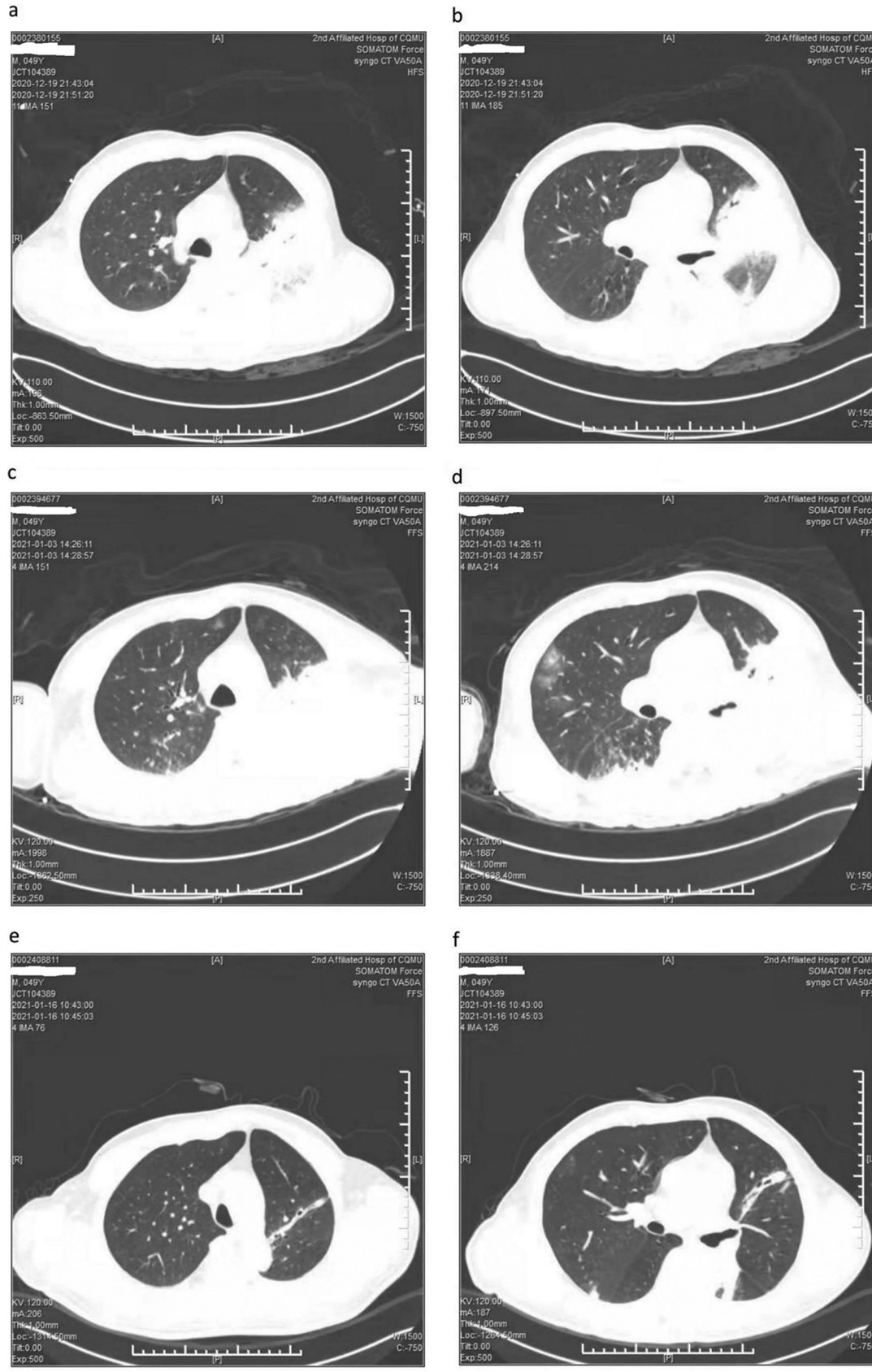

**FIG 3** CT of a 49-year-old man (the sixth patient). (a, b) The initial CT scan (2 days after the onset) showed infiltrates, reticular shadows, and consolidations with bronchograms in the lower lobe of left lung. (c, d) The follow-up CT scan (15 days after the

**TABLE 7** Treatment types and durations

| Patient no. | Treatments given (date[s] [yr-mo-day] of treatment) |
|---|---|
| 1 | Biapenem (2020-06-11 to 2020-06-24), moxifloxacin (2020-06-12 to 2020-06-24) |
| 2 | Acyclovir (2020-06-27 to 2020-06-29), epocelin (2020-06-27 to 2020-06-30), biapenem (2020-06-30 to 2020-07-03), teicoplanin (2020-07-02 to 2020-07-03), moxifloxacin (2020-07-02 to 2020-07-21) |
| 3 | Biapenem (2020-10-16 to 2020-10-16), moxifloxacin (2020-10-16 to 2020-10-20), imipenem (2020-10-16 to 2020-10-20), voriconazole (2020-10-20 to 2020-11-06), oseltamivir (2020-10-20 to 2020-10-27), roxithromycin (2020-10-22 to 2020-10-23), doxycycline (2020-10-23 to 2020-11-06), teicoplanin (2020-10-23 to 2020-11-06) |
| 4 | Biapenem (2020-11-01 to 2020-11-13), tigecycline (2020-11-01 to 2020-11-13), caspofungin (2020-11-03 to 2020-11-13) |
| 5 | Biapenem (2020-12-11 to 2020-12-22), doxycycline (2020-12-17 to 2020-12-22) |
| 6 | meropenem (2020-12-19 to 2020-12-23), moxifloxacin (2020-12-20 to 2020-12-31), imipenem (2020-12-24 to 2020-12-26, 2020-12-30 to 2020-01-15), minocycline (2020-12-26 to 2020-12-29), doxycycline (2020-12-29 to 2021-01-16), tigecycline (2020-12-31 to 2021-01-15), piperacillin (2021-01-15 to 2021-01-22) |
| 7 | Biapenem (2021-01-10 to 2021-01-11), imipenem (2021-01-12 to 2021-01-25), voriconazole (2021-01-12 to 2021-01-13), teicoplanin (2021-01-12 to 2021-01-14), doxycycline (2021-01-13 to 2021-01-25), acyclovir (2021-01-21 to 2021-01-25), cefoperazone sodium-sulbactam sodium (2021-01-26 to 2021-02-05), fluconazole (2021-01-13 to 2021-02-05) |
| 8 | Meropenem (2021-01-24 to 2021-01-25), levofloxacin (2021-01-25 to 2021-02-07), piperacillin (2021-01-25 to 2021-01-28), moxifloxacin (2021-02-07 to 2021-02-21) |
| 9 | Piperacillin (2020-12-24 to 2020-12-28), minocycline (2020-12-24 to 2020-12-28) |
| 10 | Moxifloxacin (2020-12-23 to 2020-12-24), biapenem (2020-12-24 to 2020-12-31), minocycline (2020-12-24 to 2020-12-31) |
| 11 | Moxifloxacin (2020-12-28 to 2021-01-07), minocycline (2020-12-30 to 2021-01-07) |
| 12 | Biapenem (2020-10-30 to 2020-11-09), oseltamivir (2020-10-30 to 2020-11-03), moxifloxacin (2020-10-30 to 2020-11-08) |
| 13 | Biapenem (2021-05-13 to 2021-05-16), moxifloxacin (2021-05-16 to 2021-05-20), minocycline (2021-05-16 to 2021-05-20) |

**Outcomes.** After the correct treatment of psittacosis pneumonia was initiated, patients' clinical symptoms gradually improved, and various inflammatory indicators dropped sharply to near normal. Consolidations and infiltrates on CT images were absorbed, and all patients recovered and were discharged.

## DISCUSSION

In this retrospective study, 13 patients were all infected with *Chlamydia psittaci*, and they were all diagnosed by mNGS. Typical symptoms of psittacosis include malaise, fever, mild cough, headache, and myalgia (2). These patients' clinical features were not absolutely the same, but they had many similar symptoms. Almost all patients had fever and weakness during the progression of this disease. Some also had cough and poor appetite. Among these patients, there was a patient with headache and fever as the main symptoms at the onset of disease, which is rare in psittacosis patients. Reports of psittacosis pneumonia with headache are also rare. In Chen's literature on psittacosis, there were also patients with prominent clinical manifestations of headache, and one patient was even suspected of having meningitis (9). This clinical manifestation can easily be misdiagnosed as an infection of the intracranial system or other neurological diseases, thus delaying diagnosis and treatment. This also reminds us that in clinical practice, if we encounter headache that cannot be explained by neurological disease, we can further examine to rule out pulmonary infection. There are no characteristic clinical manifestations and images of psittacosis pneumonia which can allow us to differentiate it from other pathogens causing CAP. CAP caused by psittacosis has a higher rate of leading to severe pneumonia; however, even severe psittacosis pneumonia patients can be cured if they are treated correctly and promptly. This will tell us more about psittacosis pneumonia. White blood cell (WBC) count is usually normal to slightly lowered during the acute phase of the disease (14), but in our patients, we could see an increase in WBC. The reason may be that some patients had psittacosis pneumonia complicated with other diseases, or they were not in the acute phase of the disease.

**FIG 3** Legend (Continued)
onset) showed there were still infiltrates, reticular shadows, and consolidations with bronchograms in the lower lobe of left lung. (e, f) The follow-up CT scan (28 days after the onset) showed the area of infiltrates and consolidations in the lower lobe of left lung had disappeared.

Some patients had abnormal liver function, which is consistent with previous research (15), but the abnormal liver function is not related to the basic disease of hepatitis B. After the corresponding treatment, their liver function has recovered, so we wonder whether it was hepatophilic or may be related to the severity of infection, for which more research is needed. All patients had elevated plasma D-dimer levels, but they did not have thrombosis, and some patients showed a decrease when they were retested after treatment, which may be related to inflammatory response. It cannot be ruled out that psittacosis infection may lead to an increased risk of thrombosis, and more studies are needed to confirm this.

What should be of concern is that all patients have contact history. One patient had contact with cats, and all others had contact with birds or poultry. Some once went to a raw poultry market; some raised chickens and ducks. Two patients only touched the innards of killed poultry before cooking in the kitchen, which may be an atypical exposure history that was not reported before, to our knowledge. What's more, there was a family aggregation onset in this retrospective study; three patients are from the same family, and they all had contact with sick parrots and were in charge of breeding and taking care of the sick parrots. But the disease did not transmit from person to person, as it developed only in the family members who had close contact with the dead parrots, but not in other family members. One study about occurrence and prevalence of *Chlamydia psittaci* in racing pigeons shows that humans who come in repeated close contact with the birds are at high risk of acquiring a potentially lethal zoonotic infection (16), so the contact history is very important.

The treatment of psittacosis pneumonia is clear. Tetracyclines, macrolides, and quinolones can be used to treat *C. psittaci*; they can interfere with DNA and protein synthesis (17). Tetracyclines are often used to treat psittacosis, such as doxycycline (100 mg peroral [p.o.] every 12h [q12h]) or tetracycline hydrochloride (500 mg p.o. q6h). In order to be effective and prevent relapse, treatment must continue for at least 10 to 14 days (18, 19). In this study, all patients were treated with tetracyclines or quinolones. All of the remaining patients were treated with tetracyclines and cured. Therefore, we speculate that tetracycline therapy may be superior to quinolone therapy for psittacosis pneumonia, and more experimental data are needed to confirm this.

Through mNGS testing, we found that some patients were also infected by other pathogens. The interpretation of all mNGS results should not only be based on read cutoffs alone but also combined with the clinic. We need to check whether clinical manifestations, other laboratory test results and imaging changes are consistent with the pathogens detected by mNGS. If a large number of background bacteria or miscellaneous bacteria sequences are present without dominant microorganisms, contamination should be considered first, followed by opportunistic pathogens (20). *Tropheryma whipplei* is a common commensal organism, and *Haemophilus parainfluenzae* is a common respiratory colonization bacterium (21). *Lactobacillus gasseri* and *Lactobacillus salivarius* are common oral colonization bacteria, especially when the sample for mNGS is sputum. For these bacteria, we generally do not consider them pathogenic bacteria and do not give special treatment for them. When determining opportunistic pathogens by considering etiology, the immune status of patients, underlying diseases, and source of specimens should be considered (20). For example, the mNGS report of patient 3 suggested *Candida glabrata* and virus infection. Considering his serious condition, low immune status, symptom of muscle pain, and that his (1-3)-$\beta$-D-glucan index was 109.8 pg/mL, which was much higher than normal, we considered that his case was complicated with fungal and viral infection, and we added voriconazole and oseltamivir to his treatment. The mNGS results of patient 4 also showed *Candida albicans* infection. Combined with the fact that he was seriously ill and sputum culture also suggested yeast infection, antifungal treatment was added with caspofungin. As for patient 10, her sputum mNGS results also showed *Candida albicans* infection, but her blood mNGS did not show

corresponding fungi. Combined with the fact that she was not seriously ill, the specimen for mNGS was sputum, and other fungus-related tests were negative, we considered it an oral contamination, so we did not use antifungal drugs, and the patient also recovered. The mNGS result of patient 13 also reported *Candida albicans*, but his condition was relatively mild, he was in good physical condition, and other fungi-related tests were negative, so we did not consider it etiology and did not add the corresponding drugs for fungal infection; he also recovered without antifungal therapy. For patient 12, mNGS results suggested virus infection, she had general pain and fatigue, and the blood test also showed influenza virus type B positive, so an antiviral drug was added.

The diagnosis of psittacosis pneumonia is not easy since the symptoms are not unique and can also be seen in other diseases, and not all the patients can be easily reached for their contact history. Diagnosis in patients with pneumonia usually does not include tests for *C. psittaci* infection, so many psittacosis index patients are discovered late. Some studies use PCR for diagnosis (6, 22). PCR gene expansion detection has become the auxiliary detection of molecular biological diagnosis due to its high sensitivity and specificity, but PCR testing is not Clinical Laboratory Improvement Amendments (CLIA) validated currently (23). PCR is only performed if clinicians suspect *C. psittaci* infection (9). mNGS is an emerging test technology that has emerged in recent years. The advantage of mNGS is that it has a wide detection range and does not need to specify the suspected causative microorganism prior to testing, and it can be used in the diagnosis of encephalitis, meningitis, and lower respiratory tract infections. mNGS is a culture-independent technology and can be used to detect pathogens that cannot be detected by traditional methods (23). In our hospital, mNGS results can be obtained in 48 to 72 h, but routine sputum culture takes 5 to 7 days, and many cultures are negative. The ability of mNGS to obtain timely and precise microbial diagnoses of infections is a key advantage. However, the cost of mNGS is relatively high in clinic, so it is not routinely performed in all cases of CAP. When treating patients with pneumonia of unknown or rare pathogen and severe pneumonia, physicians need to identify the causative pathogen as early as possible and make an accurate diagnosis to provide targeted treatment. Under these conditions, we will use mNGS, especially after we have already screened for common pathogens by other conventional testing methods but received positive results or the positive results could not explain the disease. Because of atypical clinical features and diagnostic challenges, psittacosis is often misdiagnosed. It is worthwhile to use mNGS in patients with pneumonia to minimize the time to diagnosis of psittacosis and the course of the disease.

This retrospective study has some limitations. First, the number of patients is small, but this is because psittacosis pneumonia is rarely seen. Second, some patients had not only psittacosis infection but also some other pathogen infections at the same time, which may affect some symptoms of the patients and disturb our judgment. On the other hand, this also showed the advantages of mNGS. mNGS can quickly and objectively detect increasing numbers of pathogenic microorganisms in clinical specimens in one test without the need for specific amplification and has an advantage in the diagnosis of rare pathogenic bacteria in difficult cases, such as psittacosis pneumonia. It can also help us to identify the pathogens in the case of multipathogenic infection to prevent misdiagnosis.

## MATERIALS AND METHODS

**Study design.** We conducted a retrospective study of 13 patients with psittacosis pneumonia who were admitted to the Second Affiliated Hospital of Chongqing Medical University between June 2020 and May 2021. We extracted these cases from electronic medical records; they were conducted with data on symptoms, clinical features, laboratory data, computed tomography, treatment, response to treatment, and outcomes of the disease.

The study protocol was approved by the Ethics Committee of the Second Affiliated Hospital of Chongqing Medical University and conducted in compliance with the Declaration of Helsinki. All data were anonymized prior to analysis.

**Methods.** mNGS was conducted using the following operational steps (24–26).

**(i) Sample processing and DNA extraction.** Clinical samples (sputum or bronchoalveolar lavage fluid [BALF]) were collected by following the standards of aseptic processing procedures and treated with enzymes for liquefaction within 24 to 48 h from collection at 4°C. Up to 600-$\mu$L samples were

transferred to new sterile tubes. Cell walls of microbes were broken by vortex with glass beads following the described conditions. DNA was extracted from 300 $\mu$L treated sputum or BALF using the TIANamp micro DNA kit (DP316; Tiangen Biotech, Beijing, China) following the manufacturer's operation manual. The extracted DNA specimens were used for the construction of DNA libraries.

For clinical samples (blood), 3- to 4-mL blood samples were collected from patients, placed in cell-free blood collecting tubes, stored at room temperature before plasma separation, and then centrifuged at 1,600 $\times$ *g* for 10 min at 4°C. Plasma samples were transferred to new sterile tubes. DNA was extracted from 300 $\mu$L of plasma using the TIANamp micro DNA kit following the manufacturer's operation manual. The extracted DNA specimens were used for the construction of DNA libraries.

**(ii) Controls.** The internal control, named UMSI (unique molecular spiked-in), was added to the sample before the DNA extraction. The sequence of UMSI varied in different samples. Each mNGS assay run included an external negative control that ran in parallel with clinical samples. During analysis, the contamination between samples could be found if the UMSI sequence was the same or the reads of some pathogens in the external control were very high.

**(iii) Library construction.** DNA libraries were constructed through transposase-mediated methods (Vision Medicals, China). An Agilent 2100 bioanalyzer was used for quality control of the DNA libraries. The quality of the DNA libraries was assessed using a Qsep1 biofragment analyzer (BiOptic Inc., La Canada Flintridge, CA) to measure the adapters and the sizes of fragments before sequencing. The size of the qualified library was ~300 to 500 bp without adapters and PCR dimers, and the concentration of the library was greater than 0.5 ng/$\mu$L. Finally, qualified DNA libraries were pooled and sequenced on the NextSeq 550Dx sequencing platform (Illumina, San Diego, CA) and BGISeq-50/MGISeq-2000 platform. Approximately 20 to 25 libraries were sequenced on a run, with a read length of 75 bp and 20 M reads per sample.

**(iv) Data analysis.** High-quality sequencing data were generated by removing low-quality and short (length < 40 bp) reads, followed by computational subtractions of human host sequences mapped to the human reference genome (hg19, hg38, and YH sequences) using Burrows-Wheeler alignment (BWA-0.7.17 [r1188]). The remaining data, by removal of low-complexity reads, were classified by simultaneously aligning to four microbial genome databases consisting of viruses, bacteria, fungi, and parasites. The reference database was constructed through data collection, data cleaning, and big data training. For the data collection, genomes were downloaded from an open database such as NCBI, China National GenBank Database (CNGBdb), or EMBL or the sequenced clinical strains. For the data cleaning, taxonomy, sequencing quality, assembly quality, and genomic completeness were assessed. For the big data training, according to the clinical sequencing data and case reports, the epidemic strains were selected. The cutoff for detection of an atypical pathogen like *C. psittaci* is one read, while for other pathogens, the cutoff is three reads.

In the end, the multiple parameters of species in microbial genome databases were calculated and exported, and professionals with microbiology and clinical background conducted interpretations of the results.

**Diagnostic criteria for psittacosis pneumonia.** To be included in the review, patients diagnosed with psittacosis pneumonia had to fulfill the following criteria: (i) meet the criteria for community-acquired pneumonia. In addition to a constellation of suggestive clinical features, a demonstrable infiltrate by chest radiograph or other imaging technique, with or without supporting microbiological data, is required for the diagnosis of pneumonia (13); and (ii) have specific fragments of DNA of *C. psittaci* identified using mNGS.

Diagnosis of severe pneumonia and ARDS had to meet the criteria for severe community-acquired pneumonia (13), which includes either one major criterion or three or more minor criteria. Minor criteria include respiratory rate of $\geq$30 breaths/min, PaO$_2$/FiO$_2$ ratio of $\leq$250, multilobar infiltrates, confusion/disorientation, uremia (blood urea nitrogen level $\geq$ 20 mg/dL), leukopenia (white blood cell count < 4,000 cells/$\mu$L), thrombocytopenia (platelet count < 100,000/$\mu$L), hypothermia (core temperature < 36°C), and hypotension requiring aggressive fluid resuscitation. Major criteria include septic shock with need for vasopressors and respiratory failure requiring mechanical ventilation (13).

The severity of ARDS is classified according to the degree of hypoxemia (PaO$_2$/FiO$_2$ ratio), with mutually exclusive categories of mild (PaO$_2$/FiO$_2$ ratio, 201:300), moderate (PaO$_2$/FiO$_2$ ratio, 101: 200), and severe (PaO$_2$/FiO$_2$ ratio $\leq$ 100) (27).

**Data availability.** We have uploaded the specific experimental data about mNGS to BioProject under accession no. PRJNA791681.

## SUPPLEMENTAL MATERIAL

Supplemental material is available online only.
**SUPPLEMENTAL FILE 1**, PDF file, 0.7 MB.

## ACKNOWLEDGMENTS

We received financial support from the Senior Medical Talents Program of Chongqing for Young and Middle-Aged, CQMU Program for Youth Innovation in Future Medicine (grant number W0118) and Kuanren Talents Program of the Second Affiliated Hospital of Chongqing Medical University (grant number KY2019Y003).

We report no potential conflict of interest.

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
