## [Reviewer comments · Microbiology Spectrum]

Microbiology Spectrum

The application of metagenomic next-generation sequencing in the diagnosis of pneumonia caused by *Chlamydia psittaci*

Junli Tang, Wanmei Tan, Lingxin Luo, Huan X, and Na Li

Corresponding Author(s): Na Li, Second Affiliated Hospital of Chongqing Medical University

Review Timeline:

Submission Date:	January 5, 2022
Editorial Decision:	February 22, 2022
Revision Received:	March 25, 2022
Editorial Decision:	April 22, 2022
Revision Received:	May 3, 2022
Editorial Decision:	May 24, 2022
Revision Received:	May 25, 2022
Accepted:	June 6, 2022

Editor: Hui Wang

Reviewer(s): Disclosure of reviewer identity is with reference to reviewer comments included in decision letter(s). The following individuals involved in review of your submission have agreed to reveal their identity: Peng Li (Reviewer #2)

Transaction Report:

DOI: <https://doi.org/10.1128/spectrum.02384-21>

February 22, 2022

Prof. Na Li
Second Affiliated Hospital of Chongqing Medical University
chongqing
China

Re: Spectrum02384-21 (The application of metagenomic next-generation sequencing in the diagnosis of pneumonia caused by *Chlamydia psittaci*)

Dear Prof. Na Li:

Link Not Available

Sincerely,

Hui Wang

Journals Department
Reviewer comments:

Reviewer #1 (Comments for the Author):

This manuscript presents a case review of 13 patients diagnosed with psittacosis pneumonia by mNGS. While overall the results and conclusions are presented relatively clearly, some missing details around the mNGS experimental design and data analysis reduce my enthusiasm for this manuscript. In particular, the apparent lack of basic positive/negative controls during the mNGS workflow is a major concern, particularly when numerous "pathogens" are being reported for each sample.

Major comments:

-Importance - add discussion of the impact of mNGS-based diagnostics for unusual diagnoses such as psittacosis pneumonia

- Introduction:
- What is the geographical distribution of *C. psittaci*?
- Lines 80-87: There are many recent papers that describe the use of mNGS for clinical diagnostics, e.g., from Charles Chiu at UCSF, and specifically in the case of pneumonia (including psittacosis pneumonia). Relevant papers should be cited here.
- Methods:
- Please provide detail on controls included during sample processing for mNGS. Were positive or negative controls included alongside clinical samples? How was contamination identified during analysis?
- Were all patients tested with both blood and respiratory samples? Why or why not?
- Has the mNGS assay used been previously validated for diagnostic use? Is mNGS routinely performed on cases of CAP?
- Lines 138-141 - How many libraries were sequenced on a run, and what was the read length and target read depth?
- Lines 146-149 - How was this alignment performed? What software was used for alignment?
- Line 151-154 - Please provide additional specific detail on how the reference databases were curated - what does "multi-parameters of species in Microbial Genome Databases were calculated" mean?
- Line 162: What was the cutoff for detection of *C. psittaci* (or any other pathogen)? One read or more?
- Results:
- Line 224 and 278- Please address the clinical significance of other pathogens detected by mNGS. Were any considered contaminants? If so, why?
- Line 245-261 - Were these patients tested for *C. psittaci* using an orthogonal method to confirm the diagnosis?
- Line 279 - Please expand discussion on the other pathogen diagnoses in this patient group - what other diagnoses were made, and what treatments were given to patients diagnosed with additional pathogens by mNGS?
- Formatting of tables is difficult to read.

Minor comments:

- Line 64: "...*C. psittaci* is hardly to grow" - does this mean it is difficult to grow, or it grows slowly in culture?
- Methods: line 146 - Please provide version information for bwa
- Results: line 223, should "chlamydia parrots" be "*C. psittaci*"?
- Line 319-320: "Which will tell us more about psittacosis pneumonia." - this is not a complete sentence
- line 325: This should be changed to "Katsura et al." or "previous research" or "previous reports"
- lines 359-361 - Move to results section
- line 374: OK to abbreviate CLIA
- line 391-394: This sentence is difficult to interpret
- Ensure *C. psittaci* is spelled correctly, capitalized correctly, and italicized throughout the manuscript

Staff Comments:

Preparing Revision Guidelines

Please return the manuscript within 60 days; if you cannot complete the modification within this time period, please contact me. If you do not wish to modify the manuscript and prefer to submit it to another journal, please notify me of your decision immediately so that the manuscript may be formally withdrawn from consideration by Microbiology Spectrum.

If your manuscript is accepted for publication, you will be contacted separately about payment when the proofs are issued; please follow the instructions in that e-mail. Arrangements for payment must be made before your article is published. For a

complete list of **Publication Fees**, including supplemental material costs, please visit our website.

This manuscript presents a case review of 13 patients diagnosed with psittacosis pneumonia by mNGS. While overall the results and conclusions are presented relatively clearly, some missing details around the mNGS experimental design and data analysis reduce my enthusiasm for this manuscript. In particular, the apparent lack of basic positive/negative controls during the mNGS workflow is a major concern, particularly when numerous "pathogens" are being reported for each sample.

Major comments:

- Importance - add discussion of the impact of mNGS-based diagnostics for unusual diagnoses such as psittacosis pneumonia
- Introduction:
 - What is the geographical distribution of *C. psittaci*?
 - Lines 80-87: There are many recent papers that describe the use of mNGS for clinical diagnostics, e.g., from Charles Chiu at UCSF, and specifically in the case of pneumonia (including psittacosis pneumonia). Relevant papers should be cited here.
- Methods:
 - Please provide detail on controls included during sample processing for mNGS. Were positive or negative controls included alongside clinical samples? How was contamination identified during analysis?
 - Were all patients tested with both blood and respiratory samples? Why or why not?
 - Has the mNGS assay used been previously validated for diagnostic use? Is mNGS routinely performed on cases of CAP?
 - Lines 138-141 - How many libraries were sequenced on a run, and what was the read length and target read depth?
 - Lines 146-149 - How was this alignment performed? What software was used for alignment?
 - Line 151-154 - Please provide additional specific detail on how the reference databases were curated - what does "multi-parameters of species in Microbial Genome Databases were calculated" mean?
 - Line 162: What was the cutoff for detection of *C. psittaci* (or any other pathogen)? One read or more?
- Results:
 - Line 224 and 278- Please address the clinical significance of other pathogens detected by mNGS. Were any considered contaminants? If so, why?
 - Line 245-261 - Were these patients tested for *C. psittaci* using an orthogonal method to confirm the diagnosis?
 - Line 279 - Please expand discussion on the other pathogen diagnoses in this patient group - what other diagnoses were made, and what treatments were given to patients diagnosed with additional pathogens by mNGS?
- Formatting of tables is difficult to read.

Minor comments:

- Line 64: "...*C. psittaci* is hardly to grow" - does this mean it is difficult to grow, or it grows slowly in culture?
- Methods: line 146 - Please provide version information for bwa
- Results: line 223, should "chlamydia parrots" be "*C. psittaci*"?

- Line 319-320: "Which will tell us more about psittacosis pneumonia." - this is not a complete sentence
- line 325: This should be changed to "Katsura et al." or "previous research" or "previous reports"
- lines 359-361 - Move to results section
- line 374: OK to abbreviate CLIA
- line 391-394: This sentence is difficult to interpret
- Ensure *C. psittaci* is spelled correctly, capitalized correctly, and italicized throughout the manuscript

Response to Reviewers

Re: Manuscript ID: Spectrum02384-21, and Title: "The application of metagenomic next-generation sequencing in the diagnosis of pneumonia caused by *Chlamydia psittaci*."

Dear editor and reviewers,

Thanks for your letter and for the reviewers' comments concerning our manuscript entitled "**The application of metagenomic next-generation sequencing in the diagnosis of pneumonia caused by *Chlamydia psittaci*.**" (Manuscript ID: **Spectrum02384-21**). We appreciate a lot for the work of the reviewer and the precious comments to our manuscript. These comments are all valuable and very helpful for revising and improving our paper, as well as the important guiding significance to our researches. We have studied all the comments carefully and have made corrections which we hope meet with approval. Revised portions are marked in red in the paper. The main corrections in the paper and the responses to the reviewer's comments are as follows:

Responses to the reviewer's comments:

1-Reviewer

Major comments:

-Importance - add discussion of the impact of mNGS-based diagnostics

for unusual diagnoses such as psittacosis pneumonia

Response: Thanks for your professional suggestion and sorry for our negligence of not discussing the impact of mNGS-based diagnostics for unusual diagnoses such as psittacosis pneumonia.

We have added the corresponding sentences in the revised draft: “mNGS can quickly and objectively detect more and more rare pathogenic microorganisms in clinical specimens, without the need for specific amplification, and has an advantage in the diagnosis of rare pathogenic bacteria in difficult cases, such as psittacosis pneumonia. The use of mNGS can improve the accuracy and reduce the delay in the diagnosis of psittacosis, which shortens the course of the disease control.” in Line 52 in the original manuscript, in Line 53 in the revised manuscript.

-Introduction:

-What is the geographical distribution of *C. psittaci*?

Response: Thank you for your professional remind to make the introduction of the psittacosis pathogen more comprehensive. We have added the corresponding sentence “*C. Psittaci* is traditionally regarded as an avian pathogen of global distribution [2].” in the revised draft in Line 60 in the original manuscript, in Line 67 in the revised manuscript.

-Lines 80-87: There are many recent papers that describe the use of

mNGS for clinical diagnostics, e.g., from Charles Chiu at UCSF, and specifically in the case of pneumonia (including psittacosis pneumonia).

Relevant papers should be cited here.

Response: Thanks very much for your suggestion. This article is indeed a very detailed introduction to mNGS, and it is a rare and excellent work, which enables us to know more about mNGS. We have added the corresponding sentence “Recently, there are more and more applications of mNGS in clinical diagnosis. It can help the diagnosis of pathogenic microbe infections in various systems, such as nervous system, gastrointestinal system, respiratory system (including psittacosis pneumonia), etc [9,10].” in the revised draft in Line 87 in the original manuscript, in Line 96 in the revised manuscript.

-Methods:

-Please provide detail on controls included during sample processing for mNGS. Were positive or negative controls included alongside clinical samples? How was contamination identified during analysis?

Response: Sorry for not describing quality control in detail in the article. The internal control, named UMSI (Unique-Molecular-Spiked-In), was added to the sample before the DNA extraction. The sequence of UMSI was different in the different sample. And each mNGS assay run included an external negative control that were run in parallel with clinical samples.

During analysis, the contamination between samples could be found if the UMSI sequence was the same or the reads of some pathogens in the external control was very high.

-Were all patients tested with both blood and respiratory samples? Why or why not?

Response: No, not all patients tested with both blood and respiratory samples. For diagnosis, bronchoalveolar lavage fluid (BALF) can exclude the possibility of sample contamination better because it extracts sputum from the bronchi and reflects the real pathogenic bacterial infection of the lungs. However, bronchoalveolar lavage requires bronchoscopy, but some patients are too old or too seriously ill to tolerate bronchoscopy, and some patients were unwilling to complete bronchoscopy due to their own reasons. So for these patients, we chose their sputum or blood as testing specimens. Thanks for your professional question.

-Has the mNGS assay used been previously validated for diagnostic use?

Response: Yes, it has used been previously validated for diagnostic use. Metagenomic next-generation sequencing (mNGS) has emerged as a promising single, universal pathogen detection method for infectious disease diagnostics [1]. mNGS had a higher sensitivity than the traditional method, especially in blood, bronchoalveolar lavage fluid and

sputum samples [2]. It is used to diagnose infections of the respiratory system[3], blood system[4], nervous system[5] and other systems[6].

[1]Simner PJ, Miller S, Carroll KC. Understanding the Promises and Hurdles of Metagenomic Next-Generation Sequencing as a Diagnostic Tool for Infectious Diseases. Clin Infect Dis. 2018;66(5):778-788. doi:10.1093/cid/cix881

[2]Duan H, Li X, Mei A, et al. The diagnostic value of metagenomic next-generation sequencing in infectious diseases. BMC Infect Dis. 2021;21(1):62. Published 2021 Jan 13. doi:10.1186/s12879-020-05746-5

[3]Qian YY, Wang HY, Zhou Y, et al. Improving Pulmonary Infection Diagnosis with Metagenomic Next Generation Sequencing. Front Cell Infect Microbiol. 2021;10:567615. Published 2021 Jan 26. doi:10.3389/fcimb.2020.567615

[4]Greninger AL, Naccache SN. Metagenomics to Assist in the Diagnosis of Bloodstream Infection. J Appl Lab Med. 2019;3(4):643-653. doi:10.1373/jalm.2018.026120

[5]Miller S, Naccache SN, Samayoa E, et al. Laboratory validation of a clinical metagenomic sequencing assay for pathogen detection in cerebrospinal fluid. Genome Res. 2019;29(5):831-842. doi:10.1101/gr.238170.118

[6]Fang X, Cai Y, Mei J, et al. Optimizing culture methods according to preoperative mNGS results can improve joint infection diagnosis. Bone Joint J. 2021;103-B(1):39-45. doi:10.1302/0301-620X.103B1.BJJ-2020-0771.R2

-Is mNGS routinely performed on cases of CAP?

Response: Thanks for your professional question and remind. We have added the corresponding sentence “ mNGS is an emerging test technology that has emerged in recent years. The advantage of mNGS is that it has a wide detection range and does not need to specify the suspected causative microorganism priorly, and can be used in the diagnosis of encephalitis, meningitis, and lower respiratory tract infections. mNGS is a culture-independent technology, and can be used to detect pathogens that cannot be detected by traditional methods[24]. In our hospital, mNGS results can be obtained within 48–72 h, but routine

sputum culture takes 5–7 days, and many cultures are negative. The ability of mNGS to obtain timely and precise microbial diagnoses of infections is a key advantage. However, the cost of mNGS is relatively high in clinic, so it is not routinely performed on all cases of CAP. When treating patients with pneumonia of unknown or rare pathogen and severe pneumonia, physicians need to identify the causative pathogen as early as possible, and to make the accurate diagnosis to provide targeted treatment. In these conditions, we will use mNGS, especially after we have already screened common pathogens by other conventional testing methods but get none positive results or the positive results couldn't explain the disease. Because of atypical clinical features and the diagnostic challenges, psittacosis is often misdiagnosed. It is worthwhile to use mNGS in patients with pneumonia to minimize the time to diagnosis of psittacosis and the course of the disease.” in the revised draft in Line 376 in the original manuscript, in Line 402 in the revised manuscript.

-Lines 138-141 - How many libraries were sequenced on a run, and what was the read length and target read depth?

Response: Thanks for your kind remind. 20~25 libraries were sequenced on a run with read length of 75bp and 20M reads per sample. We have added this sentence into the revised manuscript in Line 141 in the original manuscript, in Line 154 in the revised manuscript.

-Lines 146-149 - How was this alignment performed? What software was used for alignment?

Response: Thanks for your professional question and remind. Burrows-Wheeler Alignment (BWA) was used to align the clean reads to human host sequences and the microbial reads to the microbial reference databases. Burrows-Wheeler Transform (BWT) is the data transformation algorithm of BWA.

-Line 151-154 - Please provide additional specific detail on how the reference databases were curated - what does "multi-parameters of species in Microbial Genome Databases were calculated" mean?

Response: Thanks for your professional question and remind. The reference database was constructed through data collection, data cleaning and big data training. For the data collection, genomes were downloaded from the open database, such as NCBI, CNGBdb, EMBL et al, or from the sequenced clinical strains. For the data cleaning, taxonomy, sequencing quality, assembly quality, and genomic completeness were assessed. For the big data training, according to the clinical sequencing data and case report, the epidemic strains of were selected.

“Multi-parameters of species in Microbial Genome Databases were calculated” means genus-level relative abundance, species-specific reads

number, species-level reads depth and coverage, and so on.

-Line 162: What was the cutoff for detection of *C. psittaci* (or any other pathogen)? One read or more?

Response: Thanks for your question. The cutoff for detection of atypical pathogen like *C. psittaci* is one read. While for other pathogens, the cutoff is three reads.

-Results:

-Line 224 and 278- Please address the clinical significance of other pathogens detected by mNGS. Were any considered contaminants? If so, why?

Response: Thanks for your professional question. Except *C. psittaci*, mNGS also detected other pathogens. Some of them are pathogenic bacteria such as *Haemophilus parainfluenzae*, and some are colonized bacteria such as *Lactobacillus salivarius*, *Enterococcus faecalis* etc. According to your suggestion, we have rewritten this paragraph and added corresponding explanations “*C. Psittaci* was detected by mNGS in all patients. mNGS can detect not only *C. Psittaci*, but also other microorganisms at the same time. Some of them are pathogenic bacteria such as *Haemophilus parainfluenzae* and some are colonized bacteria such as *Lactobacillus salivarius*, *Enterococcus faecalis*, etc. It was found

that less colonized bacteria were detected in BALF and blood, while a large number of colonized bacteria were detected in sputum such as case 3 and case 10. This is because sputum is discharged through the mouth and sputum specimens are susceptible to contamination by oropharyngeal colonized bacteria.” in the revised manuscript in Line 223 in the original manuscript, in Line 237 in the revised manuscript.

In patient 3, because the sample for mNGS detection was sputum, except *Chlamydia psittaci*, mNGS detected *Enterococcus faecalis*, *Tropheryma whipplei*, *Candidaglabrata*, *Human papillomavirus100*, *Torquetenovirus* and *Human gammaherpesvirus 4* which we regard as oral colonized bacteria. And we gave empirical antibiotic therapy with biapenem, quinolones and imipenem after admission which are all effective antibiotics for these pathogenic microorganism infections. At the same time, we gave antifungi therapy with voriconazole which is effective for *Candidaglabrata* infection before mNGS results returned to us. But they didn't work. After we added doxycycline which is effective for *Chlamydia psittaci* infection, the symptoms of fever, cough and general conditions improved gradually.

In patient 10, because the sample for mNGS detection was sputum, except *Chlamydia psittaci*, mNGS detected *Lactobacillus gasseri*, *Lactobacillus salivarius*, *Enterococcus faecalis*, *Tropheryma whipplei*, *Haemophilus parainfluenzae* which we regard as oral colonized bacteria.

And we gave empirical antibiotic therapy with quinolones and biapenem on admission which are all effective antibiotics for *Haemophilus parainfluenzae* infection. But they didn't work. After we added minocycline which is effective for *Chlamydia psittaci* infection, the symptoms of fever disappeared within 3 days, and the cough and general conditions improved gradually.

-Line 245-261 - Were these patients tested for *C. psittaci* using an orthogonal method to confirm the diagnosis?

Response: Thanks for your professional question.

Culture has high specificity and is often used to confirm the diagnosis. Hence, after we diagnosed psittacosis, we asked our hospital, other hospital and CCDC (Chinese Center for Disease Control and Prevention) in Chongqing and neighboring provinces and cities about culture, but they said biosafety level three is needed for *C. psittaci* culture, they don't have the qualification to do the culture. As far as we know, culture of *C. psittaci* cannot be routinely performed in most diagnostic laboratories and hospitals in China. We also tried to do serological tests of a serum sample, and asked many hospitals, laboratories and big third-party testing company in China, but they said they didn't have the reagents. So we gave up to verify by using traditional methods.

We had ever tried to use the remaining extracted DNA nucleic acid from

three patients' samples to do PCR testing and Sanger Sequencing for *C. psittaci* in laboratory. The results of PCR and Sanger Sequencing verified the existing of *C. psittaci*, and are consistent with the results of mNGS.

In patient 11 whose BALF were detected 779 specific *C. psittaci* sequences and no other pathogen by mNGS, the PCR amplification bands (No.20884) was clear (Fig. 1) and the Sanger Sequencing results have a Query Coverage of 95%, a Per. Identity of 100% and the E. Value was 0.0 by BLAST comparison with *C. psittaci* (**Appedix. 1**).

In patient 10 whose sputum were detected 379 specific *C. psittaci* sequences and some other oral colonized pathogen by mNGS, the PCR amplification bands (No.20638) was clear (Fig. 1) but the Sanger Sequencing results couldn't be verified by BLAST due to the mixture with other pathogen sequences. However, in the blood sample of patient 10 which were detected only 3 specific *C. psittaci* sequences and no other pathogen by mNGS, the PCR amplification bands (No.7262) was not found (Fig. 1).

In patient 9 whose BALF were detected only 2 specific *C. psittaci* sequences by mNGS, the PCR amplification bands (No.20707) was not found (Fig. 1) because the amount of *C. psittaci* genomic DNA sequences were too small to be amplified by PCR.

Fig 1. PCR amplification results of 3 patients' samples. No.20707 is DNA nucleic acid extracted from the BALF of patient 9 with 2 specific *C. psittaci* sequences by mNGS. No.20884 is DNA nucleic acid extracted from the BALF of patient 11 with 779 specific *C. psittaci* sequences by mNGS. No. 20638 is DNA nucleic acid extracted from the sputum of patient 10 with 379 specific *C. psittaci* sequences by mNGS. No. 7262 is DNA nucleic acid extracted from the blood of patient 10 with 3 specific *C. psittaci* sequences by mNGS. The target size of PCR product is 393bp (Opota, Onya; Vanrompay, Daisy; Greub, Gilbert; Branley, James; Longbottom, David; Erard, Veronique; Jaton, Katia; Borel, Nicole (2015). Improving the molecular diagnosis of Chlamydia psittaci and Chlamydia abortus infection with a species-specific duplex real-time PCR. Journal of Medical Microbiology, 64(10):1174-1185. DOI: <https://doi.org/10.1099/jmm.0.000139>)

Because it is a retrospective study, not all the patients had enough remaining samples to do PCR testing and Sanger Sequencing for *C. psittaci* in laboratory. But the above results of PCR and Sanger Sequencing verified the existing of *C. psittaci*, and are consistent with the results of mNGS, which can have certain representativeness.

-Line 279 - Please expand discussion on the other pathogen diagnoses in this patient group - what other diagnoses were made, and what treatments were given to patients diagnosed with additional pathogens by mNGS?

Response: Thanks for your remind. Through mNGS testing, we found that some patients were also infected by other pathogens. For these patients, we also have treatments that target other pathogens. So the treatment of them were not the same. For example, for viral infections, we have added antiviral therapy; for fungal infections, we added antifungal therapy; for bacterial infection, we apply antibiotics accordingly. (Table 7) We have added this paragraph into the revised manuscript in Line 278 in the original manuscript, in Line 300 in the revised manuscript.

-Formatting of tables is difficult to read.

Response: Sorry for the trouble caused by our form format. We have modified the form format and we hope it could be easier to be read. If

there is any problem, don't hesitate to tell us. Thanks for your kind remind. We have modified the typesetting of the Table 1/2/3 and simplified the content of the Table 6, added the contents of Table 5.

Minor comments:

-Line 64: "...*C. psittaci* is hardly to grow" - does this mean it is difficult to grow, or it grows slowly in culture?

Response: I am very sorry for the misunderstanding caused by my improper description. This means "*C. psittaci* grows slowly in culture". We have revised this sentence into "*C. Psittaci* grows slowly in culture" in the revised manuscript.

-Methods: line 146 - Please provide version information for bwa

Response: Thanks for your kind remind. The version information for bwa is BWA-0.7.17 (r1188), we have added this into the revised manuscript.

-Results: line 223, should "chlamydia parrots" be "*C. psittaci*"?

Response: Yes, it's "*C. psittaci*", Sorry for that we used another expression, we have changed it into "*C. Psittaci*" make it easier to read, thank you for your suggestion to make it more accurate.

-Line 319-320: "Which will tell us more about psittacosis pneumonia." -

this is not a complete sentence

Response: I'm really sorry for the incompleteness of the sentence due to my negligence. Thanks for your kind remind. We have changed this sentence into "This will tell us more about psittacosis pneumonia."

-line 325: This should be changed to "Katsura et al." or "previous research" or "previous reports"

Response: Sorry for the trouble caused by our inaccurate expression. Thanks so much for your kind remind. We have changed it to "previous research" .

-lines 359-361 - Move to results section

Response: Thanks for your remind. It has been moved to the treatment section in Line 282 in the original manuscript, in Line 309 in the revised manuscript.

-line 374: OK to abbreviate CLIA

Response: Thanks for your professional advice. It has been changed to CLIA in the revised manuscript.

-line 391-394: This sentence is difficult to interpret.

Response: I'm really sorry for the trouble caused by our inaccurate

expressions. We have changed this to “Second, some patients had not only psittacosis infection, but also some other pathogen infections in the meanwhile, which may affect some phenomenon of the patients and disturb our judgement. On the other hand, this also showed the advantages of mNGS. mNGS can quickly and objectively detect more and more pathogenic microorganisms in clinical specimens in one test, without the need for specific amplification, and has an advantage in the diagnosis of rare pathogenic bacteria in difficult cases, such as psittacosis pneumonia. It can also help us to identify the pathogens in the case of multiple pathogens infection, so as to prevent misdiagnosis.” in the revised manuscript. We hope it could be easier to be read. If there is any problem, don’t hesitate to tell us. Thanks for your kind remind.

-Ensure *C. psittaci* is spelled correctly, capitalized correctly, and italicized throughout the manuscript

Response: Thanks for your kind remind. It has been changed to “*C. Psittaci*” throughout the manuscript.

2-Reviewer

Major comments.

1) As to the question “I didn’t see much results of mNGS but detailed clinical data, and more results about the data volume, coverage, subtype of *C. psittaci*, the corresponding platform of each case and et al. should be provided.”

Response: I'm sorry for not describing the results of mNGS in details.

Thanks for your remind. **We have added the data volume and the corresponding platform of each case in Table 5.**

As to the coverage, for patients with relatively high DNA sequences, we drawn the Genome coverage map and the coverage of those patients were shown at the top of the Genome coverage map in Fig 1. But for patients with low DNA sequences which were too low to draw the Genome coverage map, we couldn’t see the coverage.

As to the subtype of *C. psittaci*, we have ever tried to draw the cladogram, but the DNA sequences were too low to draw. Hence, we couldn’t identify the accurate subtype of *C. psittaci*.

Sorry for unable to provide all the detailed results of mNGS due to the relatively low DNA sequences.

2)As to the question “The authors should clearly describe how they designed the study (retrospectively? Treatment after mNGS? It’s a little confusing),”

Response: I'm sorry that we didn't express this clearly enough. This is a retrospective study on psittacosis pneumonia detected via mNGS, with main focuses on the clinical manifestations, laboratory data, diagnosis and treatment, which aim to give us a better understanding of psittacosis pneumonia and the application of mNGS in the diagnosis of psittacosis pneumonia.

We wrote “We conducted a retrospective case review of thirteen patients admitted to the Second Affiliated Hospital of Chongqing Medical University, with psittacosis pneumonia between June 2020 and May 2021.” in the part of Study design in the original manuscript in Line 97, which is maybe not too obvious. We have changed this sentence into “**We conducted a retrospective study of thirteen patients admitted to the Second Affiliated Hospital of Chongqing Medical University, with psittacosis pneumonia between June 2020 and May 2021.**” in the revised manuscript. Thanks for your kind remind.

3)As to the question “on what conditions they introduced mNGS (the day between hospitalization and mNGS varied.)”

Response: Thanks for your question. We use mNGS for diagnosis when we have already screened common pathogens by other conventional testing methods but get none positive results, or the other positive pathogenic results couldn't explain the condition especially when they do not respond to conventional antibiotic treatment based on clinical experience, or when we highly suspected psittacosis infection according to the exposure history.

4) As to the question “what is the criterion for diagnosis by using mNGS (most cases had more specific reads of other pathogens which can also cause pneumonia)?”

Response: Thanks for your good and professional question.

The following is the diagnostic criteria for psittacosis pneumonia by using mNGS: (1) meet the criteria for community-acquired pneumonia; In addition to a constellation of suggestive clinical features, a demonstrable infiltrate by chest radiograph or other imaging technique, with or without supporting microbiological data, is required for the diagnosis of pneumonia [11]; (2) have specific fragment DNA of *C. psittaci* identified using mNGS. We wrote this criterion in the part of Diagnostic criteria for psittacosis pneumonia in the original manuscript.

C. Psittaci is an obligate intracellular bacterium, which must grow and reproduce in living cells both in vivo and in vitro. So the detection sensitivity and detection rate of intracellular bacteria is relatively low due to the small amount of bacteria released extracellularly into body fluids such as blood, sputum and bronchoalveolar lavage fluid. And because of the difficulty of DNA extraction and low possibility for contamination, the *C. Psittaci* was considered detected if: 1) its genus was among the top 20 with highest SDSMRN; 2) it ranked first within its genus; and 3) it had a SDSMRN>1 [1]. So in the interpretation of NGS results, even though the DNA sequences amount of intracellular bacteria was small, we should also consider it as pathogen with high possibility.

[1]Qian YY, Wang HY, Zhou Y, et al. Improving Pulmonary Infection Diagnosis with Metagenomic Next Generation Sequencing. *Front Cell Infect Microbiol.* 2021;10:567615. Published 2021 Jan 26.

doi:10.3389/fcimb.2020.567615

And the interpretation of all NGS results should be combined with clinic. We should comprehensively consider whether the clinical manifestations, other laboratory examination results, imaging changes and the response to anti-infective therapy are consistent with the pathogens detected by NGS. And the sample source for mNGS should also be considered. It was found that less contaminated bacteria were detected in BALF and blood, while a

large number of colonized bacteria were detected in sputum such as case 3 and case 10. This is because sputum is discharged through the mouth and sputum specimens are susceptible to contamination by oropharyngeal colonized bacteria.

In patient 3, because the sample for mNGS detection was sputum, except *Chlamydia psittaci*, mNGS detected *Enterococcus faecalis*, *Tropheryma whipplei*, *Candida glabrata*, *Human papillomavirus 100*, *Torquetenovirus* and *Human gamma herpesvirus 4* which we regard as oral colonized bacteria. And we gave empirical antibiotic therapy with biapenem, quinolones and imipenem after admission which are all effective antibiotics for these pathogenic microorganism infections. At the same time, we gave antifungi therapy with voriconazole which is effective for *Candida glabrata* infection before mNGS results returned to us. But they didn't work. After we added doxycycline which is effective for *Chlamydia psittaci* infection, the symptoms of fever, cough and general conditions improved gradually.

In patient 10, because the sample for mNGS detection was sputum, except *Chlamydia psittaci*, mNGS detected *Lactobacillus gasseri*, *Lactobacillus salivarius*, *Enterococcus faecalis*, *Tropheryma whipplei*, *Haemophilus parainfluenzae* which we regard as oral colonized bacteria. And we gave empirical antibiotic therapy with quinolones and biapenem on admission which are all effective antibiotics for *Haemophilus*

parainfluenzae infection. But they didn't work. After we added minocycline which is effective for *Chlamydia psittaci* infection, the symptoms of fever disappeared within 3 days, and the cough and general conditions improved gradually.

5) As to the question "The reason for choosing different platforms and the difference of these platforms on the results should be discussed."

Response: Thanks for your question. Because this is a retrospective study, so we just extracted the data of mNGS results from electronic medical records and couldn't choose the platforms by ourselves. As far as we know, different platform didn't have much effect on the results of mNGS. The main effects of platform on mNGS are raw error rate and base preference etc.

6) As to the question "how to confirm the results of mNGS as specific sequences of different pathogens were observed? Currently, the clinical symptoms and the contact with birds or other related environments does not necessarily indicated the patients were infected by *C. psittaci*."

Response: Thanks for your professional question.

As to your question " how to confirm the results of mNGS as specific sequences of different pathogens were observed.", we also once sought to do this.

Culture has high specificity and is often used to confirm the diagnosis. Hence, after we diagnosed psittacosis, we asked our hospital, other hospital and CCDC (Chinese Center for Disease Control and Prevention) in Chongqing and neighboring provinces and cities about culture, but they said biosafety level three is needed for *C. psittaci* culture, they don't have the qualification to do the culture. As far as we know, culture of *C. psittaci* cannot be routinely performed in most diagnostic laboratories and hospitals in China. We also tried to do serological tests of a serum sample, and asked many hospitals, laboratories and big third-party testing company in China, but they said they didn't have the reagents. So we gave up to verify by using traditional methods.

We had ever tried to use the remaining extracted DNA nucleic acid from three patients' samples to do PCR testing and Sanger Sequencing for *C. psittaci* in laboratory. The results of PCR and Sanger Sequencing verified the existing of *C. psittaci*, and are consistent with the results of mNGS. In patient 11 whose BALF were detected 779 specific *C. psittaci* sequences and no other pathogen by mNGS, the PCR amplification bands (No.20884) was clear (Fig. 1) and the Sanger Sequencing results have a Query Coverage of 95%, a Per. Identity of 100% and the E. Value was 0.0 by BLAST comparison with *C. psittaci* (**Appedix. 1**).

In patient 10 whose sputum were detected 379 specific *C. psittaci* sequences and some other oral colonized pathogen by mNGS, the PCR amplification bands (No.20638) was clear (Fig. 1) but the Sanger Sequencing results couldn't be verified by BLAST due to the mixture with other pathogen sequences. However, in the blood sample of patient 10 which were detected only 3 specific *C. psittaci* sequences and no other pathogen by mNGS, the PCR amplification bands (No.7262) was not found (Fig. 1).

In patient 9 whose BALF were detected only 2 specific *C. psittaci* sequences by mNGS, the PCR amplification bands (No.20707) was not found (Fig. 1) because the amount of *C. psittaci* genomic DNA sequences were too small to be amplified by PCR.

Fig 1. PCR amplification results of 3 patients' samples. No.20707 is DNA nucleic acid extracted from the BALF of patient 9 with 2 specific *C. psittaci* sequences by mNGS. No.20884 is DNA nucleic acid extracted

from the BALF of patient 11 with 779 specific *C. psittaci* sequences by mNGS. No. 20638 is DNA nucleic acid extracted from the sputum of patient 10 with 379 specific *C. psittaci* sequences by mNGS. No. 7262 is DNA nucleic acid extracted from the blood of patient 10 with 3 specific *C. psittaci* sequences by mNGS. The target size of PCR product is 393bp (Opota, Onya; Vanrompay, Daisy; Greub, Gilbert; Branley, James; Longbottom, David; Erard, Veronique; Jaton, Katia; Borel, Nicole (2015). Improving the molecular diagnosis of Chlamydia psittaci and Chlamydia abortus infection with a species-specific duplex real-time PCR. Journal of Medical Microbiology, 64(10):1174-1185. DOI: <https://doi.org/10.1099/jmm.0.000139>)

Because it is a retrospective study, not all the patients had enough remaining samples to do PCR testing and Sanger Sequencing for *C. psittaci* in laboratory. But the above results of PCR and Sanger Sequencing verified the existing of *C. psittaci*, and are consistent with the results of mNGS which can have certain representativeness.

For *C. psittaci* and other rare pathogenic bacteria, once detected, combined with the patient's condition and relevant laboratory examination, can guide the use of clinical antibiotics. *C. psittaci* is an obligate intracellular bacterium, which must grow and reproduce in living

cells both in vivo and in vitro. So the detection sensitivity and detection rate of intracellular bacteria is relatively low due to the small amount of bacteria released extracellularly into body fluids such as blood, sputum and bronchoalveolar lavage fluid. So in the interpretation of NGS results, even though the DNA sequences amount of intracellular bacteria was small, we should also consider it as pathogen with high possibility. And the interpretation of all NGS results should be combined with clinic. Whether the clinical manifestations, other laboratory examination results, imaging changes and the response to anti-infective therapy are consistent with the pathogens detected by NGS.

Minor comments.

-*C. Psittaci* should be in italic and C/P should be in capital. uL should be written as μL . There are many additional spaces through the manuscript, please delete.

Response: We are very sorry for the trouble brought to you by our careless writing. We have revised the careless writing in the revised manuscript. Thank you for your kind remind.

-Line 13. mNGS is not a new tool, please revise the sentence.

Response: Sorry for the trouble caused by our inaccurate expression. **We have deleted the word “new” in the revised manuscript.**

-Lines 22-24. It seems there were 22 patients in total. Please clarify.

Response: Sorry for the trouble caused by our inaccurate expression.

Instead of 22 people, the same person could have severe pneumonia or ARDS at the same time, or could be on a ventilator at the same time. We have changed it to “ **Ten patients developed ARDS, among which, six patients were severe pneumonia cases and needed ventilator therapy**” in the revised manuscript.

-Lines 24-26. The description should be revised.

Response: I'm really sorry for the trouble caused by our inaccurate expression. I have changed it to “**Most patients got psittacosis pneumonia during the cold season. Ten cases were sporadic, but three were family clustering.**” Thanks for your remind.

-Lines 56-57. What is the problem worth attention? Please clarify.

Response: I'm sorry that we didn't express this clearly enough. We have added “**such as its sick season, some special exposure history, the rate of severe disease and the high cure rate etc.**” in the revised manuscript. In Line 52 in the original manuscript, in Line 51 in the revised manuscript.

-Line 241-255, please summarize the results and rewritten the paragraph.

It is also important to give more evidence on some cases with few specific reads of *C. psittaci* but many reads of other pathogens as shown in Table 5. It's interesting to notice case 3 had more specific reads in the sample of sputum than most other cases with BALF samples, is this patient had a severe symptom?

Response: I'm sorry for not describing the results of mNGS detection in detail. Thanks for your kind remind. We have rewritten this paragraph and added corresponding explanations "*C. Psittaci* was detected by mNGS in all patients. mNGS can detect not only *C. Psittaci*, but also other microorganisms at the same time. Some of them are pathogenic bacteria such as *Haemophilus parainfluenzae* and some are colonized bacteria such as *Lactobacillus salivarius*, *Enterococcus faecalis*, etc. It was found that less colonized bacteria were detected in BALF and blood, while a large number of colonized bacteria were detected in sputum such as case 3 and case 10. This is because sputum is discharged through the mouth and sputum specimens are susceptible to contamination by oropharyngeal colonized bacteria." in the revised manuscript.

Case 3 was indeed more severe, so he could not tolerate bronchoscopy.

Hence we chose his sputum as the test sample.

-Line 517. Table 5 should contain the sequencing platform for each case and the data volume generated by mNGS.

Response: I'm sorry for not describing the results of mNGS detection in detail. We have refined the contents of **table 5** in the revised manuscript. Thanks for your remind.

-Lines 525-536. The legend of figure 1 had nearly the same description in the manuscript (Lines 241-255), please revise the description.

Response: I'm sorry for the uncomfortable experience brought to you by our cumbersome description. Thanks very much for your kind remind. We have simplified **the description below the figure 1**.

Special thanks to you for your professional and kind advice and comments. We hope the responses to reviewer's comments and revised manuscript could provide relatively reasonable and satisfactory replies to the comments of reviewer and reach the demand of publication.

Thanks again for your work and giving us opportunity for revising.

Sincerely,

Na Li

April 22, 2022

Prof. Na Li
Second Affiliated Hospital of Chongqing Medical University
chongqing
China

Re: Spectrum02384-21R1 (The application of metagenomic next-generation sequencing in the diagnosis of pneumonia caused by *Chlamydia psittaci*)

Dear Prof. Na Li:

Link Not Available

Sincerely,

Hui Wang

Journals Department
Reviewer comments:

Reviewer #1 (Comments for the Author):

Overall, the authors have done a thorough job responding to the reviewers' comments in their response document. However, the authors should keep in mind that responses to reviewer comments should also be incorporated into the manuscript wherever possible to benefit the reader. For example, the authors note in their response to reviewers that they included internal and negative controls on all runs, but they did not add this information into the methods section. This should be corrected to provide the information to the reader and not just the reviewers. In addition, their descriptions of orthogonal testing and their thresholds/criteria for pathogen detection in the mNGS data are crucial to include in the manuscript.

Major comments:

- Add information in responses to reviewer questions (including use of internal/external controls, reason for selection of blood/sputum/BAL samples, database curation info, read cutoffs, SDSMRN criteria) to the manuscript
- Add the extensive explanation of orthogonal testing and PCR/gel figure to the manuscript
- It is still unclear how the authors concluded that additional organisms identified in some samples (in addition to *C. psittaci*) are also pathogens that required treatment, rather than commensal organisms or other non-specific signal. Was this based on read cutoffs alone? Please provide additional information on this in the text. If possible, please also add information/references on how this specific mNGS assay in your hospital was validated for its current use.

Minor comments:

- *Chlamydia psittaci* should be abbreviated as *C. psittaci*. The authors seem to have misunderstood the comment to standardize their capitalization and changed all instances to a capital "P". This should be corrected back to "*C. psittaci*".
- Lines 247-256 - change "colonized" to "colonizing"

Staff Comments:

Preparing Revision Guidelines

Please return the manuscript within 60 days; if you cannot complete the modification within this time period, please contact me. If you do not wish to modify the manuscript and prefer to submit it to another journal, please notify me of your decision immediately so that the manuscript may be formally withdrawn from consideration by Microbiology Spectrum.

Overall, the authors have done a thorough job responding to the reviewers' comments in their response document. However, the authors should keep in mind that responses to reviewer comments should also be incorporated into the manuscript wherever possible to benefit the reader. For example, the authors note in their response to reviewers that they included internal and negative controls on all runs, but they did not add this information into the methods section. This should be corrected to provide the information to the reader and not just the reviewers. In addition, their descriptions of orthogonal testing and their thresholds/criteria for pathogen detection in the mNGS data are crucial to include in the manuscript.

Major comments:

- Add information in responses to reviewer questions (including use of internal/external controls, reason for selection of blood/sputum/BAL samples, database curation info, read cutoffs, SDSMRN criteria) to the manuscript
- Add the extensive explanation of orthogonal testing and PCR/gel figure to the manuscript
- It is still unclear how the authors concluded that additional organisms identified in some samples (in addition to *C. psittaci*) are also pathogens that required treatment, rather than commensal organisms or other non-specific signal. Was this based on read cutoffs alone? Please provide additional information on this in the text. If possible, please also add information/references on how this specific mNGS assay in your hospital was validated for its current use.

Minor comments:

- *Chlamydia psittaci* should be abbreviated as *C. psittaci*. The authors seem to have misunderstood the comment to standardize their capitalization and changed all instances to a capital "P". This should be corrected back to "*C. psittaci*".
- Lines 247-256 – change "colonized" to "colonizing"

Response to Reviewers

Re: Manuscript ID: Spectrum02384-21R2, and Title: "The application of metagenomic next-generation sequencing in the diagnosis of pneumonia caused by *Chlamydia psittaci*."

Dear editor and reviewers,

Thanks for your letter and for the reviewers' comments concerning our manuscript entitled "**The application of metagenomic next-generation sequencing in the diagnosis of pneumonia caused by *Chlamydia psittaci*.**" (Manuscript ID: **Spectrum02384-21R2**). Thank you very much for your re-reading of our revised article. We appreciate a lot for the work of the reviewer and the precious comments to our manuscript. These comments are all valuable and very helpful for revising and improving our paper, as well as the important guiding significance to our researches. We have studied all the comments carefully and have made corrections which we hope meet with approval. Revised portion are marked in red in the paper. The main corrections in the paper and the responds to the reviewer's comments are as the following:

Responds to the reviewer's comments:

Reviewer #1:

Major comments:

1. Add information in responses to reviewer questions (including use of internal/external controls, reason for selection of blood/sputum/BAL samples, database curation info, read cutoffs, SDSMRN criteria) to the manuscript

Response: Sorry for not describing quality control in detail in the article.

We have added the corresponding information, including use of internal/external controls, reason for selection of blood/sputum/BAL samples, database curation info, read cutoffs and SDSMRN criteria in the revised manuscript. The revisions are as the follows:

Use of internal/external controls:

We have added the corresponding sentences:“**2.Controls: The internal control, named UMSI (Unique-Molecular-Spiked-In), was added to the sample before the DNA extraction. The sequence of UMSI varied in different samples. And each mNGS assay run included an external negative control that run in parallel with clinical samples. During analysis, the contamination between samples could be found if the UMSI sequence was the same or the reads of some pathogens in the external control was very high.**” in Line 144-151 in the revised manuscript.

reason for selection of blood/sputum/BAL samples:

We have added the corresponding sentences: “Not all patients tested with both blood and respiratory samples. For diagnosis, bronchoalveolar lavage fluid (BALF) can exclude the possibility of sample contamination better because it extracts sputum from the bronchi and reflects the real pathogenic bacterial infection of the lungs. However, bronchoalveolar lavage requires bronchoscopy, some patients were too old or too seriously ill to tolerate bronchoscopy, and some patients unwilling to complete bronchoscopy due to their own reasons. So for these patients, we chose their sputum or blood as testing specimens. ” in Line 252-260 in the revised manuscript

database curation info:

We have added the corresponding sentences: “The reference database was constructed through data collection, data cleaning and big data training. For the data collection, genomes were downloaded from the open database, such as NCBI, CNGBdb, EMBL et al, or from the sequenced clinical strains. For the data cleaning, taxonomy, sequencing quality, assembly quality, and genomic completeness were assessed. For the big data training, according to the clinical sequencing data and case report, the epidemic strains of were selected.” in Line 171-179 in the revised manuscript.

read cutoffs:

We have added the corresponding sentence: “The cutoff for detection of atypical pathogen like *C. psittaci* is one read. While for other pathogens, the cutoff is three reads. ” in Line 179-180 in the revised manuscript.

SDSMRN criteria:

We have added the corresponding sentence: “*C. psittaci* is an obligate intracellular bacterium, which must grow and reproduce in living cells both in vivo and in vitro. So the detection sensitivity and detection rate of intracellular bacteria is relatively low due to the small amount of bacteria released extracellularly into body fluids such as blood, sputum and bronchoalveolar lavage fluid. And because of the difficulty of DNA extraction and low possibility for contamination, the *C. psittaci* was considered detected if: 1) its genus was among the top 20 with highest SDSMRN; 2) it ranked first within its genus; and 3) it had a SDSMRN>1 [28]. So in the interpretation of mNGS results, even though the DNA sequences amount of intracellular bacteria was small, we should also consider it as pathogen with high possibility.” in Line 501-512 in the revised manuscript.

2. Add the extensive explanation of orthogonal testing and PCR/gel figure to the manuscript

Response: Thanks for your professional remind and suggestion.

We have added the corresponding paragraphs in Line 289-318 in the revised manuscript.

“orthogonal testing

We had ever tried to use the remaining extracted DNA nucleic acid from three patients' samples to do PCR testing and Sanger Sequencing for *C. psittaci* in laboratory[16]. The results of PCR and Sanger Sequencing verified the existing of *C. psittaci*, and were consistent with the results of mNGS.

In patient 11 whose BALF were detected 779 specific *C. psittaci* sequences and no other pathogen by mNGS, the PCR amplification bands (No.20884) was clear (Fig 2) and the Sanger Sequencing results have a Query Coverage of 95%, a Per. Identity of 100% and the E. Value was 0.0 by BLAST comparison with *C. psittaci* (Appedix. 1).

In patient 10 whose sputum were detected 379 specific *C. psittaci* sequences and some other oral coloning pathogen by mNGS, the PCR amplification bands (No.20638) was clear (Fig 2) but the Sanger Sequencing results couldn't be verified by BLAST due to the mixture

with other pathogen sequences. However, in the blood sample of patient 10 which were detected only 3 specific *C. psittaci* sequences and no other pathogen by mNGS, the PCR amplification bands (No.7262) was not found (Fig 2).

In patient 9 whose BALF were detected only 2 specific *C. psittaci* sequences by mNGS, the PCR amplification bands (No.20707) was not found (Fig 2) because the amount of *C. psittaci* genomic DNA sequences were too small to be amplified by PCR.

Because it is a retrospective study, not all the patients had enough remaining samples to do PCR testing and Sanger Sequencing for *C. psittaci* in laboratory. But the above results of PCR and Sanger Sequencing verified the existing of *C. psittaci*, and are consistent with the results of mNGS which can have certain representativeness.”

3. It is still unclear how the authors concluded that additional organisms identified in some samples (in addition to *C. psittaci*) are also pathogens that required treatment, rather than commensal organisms or other non-specific signal. Was this based on read cutoffs alone? Please provide additional information on this in the text. If possible, please also add information/references on how this specific mNGS assay in your hospital was validated for its current use.

Response: Thanks for your good and professional question. The interpretation of all mNGS results should not only base on read cutoffs alone, but also combine with clinic. We need to check whether the clinical manifestations, other laboratory test results and imaging changes are consistent with the pathogens detected by mNGS. We have added the corresponding sentences “Through mNGS testing, we found that some patients were also infected by other pathogens. The interpretation of all mNGS results should not only base on read cutoffs alone, but also combine with clinic. We need to check whether the clinical manifestations, other laboratory test results and imaging changes are consistent with the pathogens detected by mNGS. If a large number of background bacteria or miscellaneous bacteria sequences are present without dominant microorganisms, contamination should be considered first, followed by opportunistic pathogens [23]. *Tropheryma whipplei* is common commensal organism and *haemophilus parainfluenzae* is common respiratory colonization bacteria [24]. *Lactobacillus gasseri*, *Lactobacillus salivarius* are common oral colonization bacteria especially when the sample of mNGS is sputum. For these bacteria, we generally do not consider them as pathogenic bacteria, and do not give special treatment for them. When determining opportunistic pathogens as etiology, the immune status of patients, underlying diseases and source of specimens should be considered[23]. For example, the mNGS report of

patient 3 suggested *Candida glabrata* and virus infection. Considering his serious condition, low immune status, the symptom of muscle pain, and his (1-3)- β -d-glucan index was 109.8 pg/ml which was much higher than normal, we considered that he was complicated with fungal and viral infection, and added voriconazole and oseltamivir in his treatment. The mNGS results of patient 4 also reported *Candida albicans* infection. Combined with the fact that he was seriously ill and sputum culture also suggested yeast infection, anti-fungal treatment was added with Capofungin. As for patient 10, her sputum mNGS results also reported *Candida albicans* infection, but her blood mNGS did not have corresponding fungi, combining with the fact that she was not seriously ill, the specimen for mNGS was sputum and other fungi-related tests were negative, we considered it as oral contamination, so we did not use anti-fungal drugs and the patient also recovered. The mNGS result of patient 13 also reported *Candida albicans*, but his condition was relatively mild and he was in good physical condition, and other fungi-related tests were negative, so we did not consider it as etiology and did not add the corresponding drugs for fungal infection. And the patient 13 also recovered without anti-fungal therapy. The patient 12, mNGS results suggested virus infection, combined with her general pain and fatigue, and the blood test also showed influenza virus TYPE B positive, so antiviral drug was added.”in Line 445-484 in the revised manuscript.

Minor comments:

1. Chlamydia psittaci should be abbreviated as C. psittaci. The authors seem to have misunderstood the comment to standardize their capitalization and changed all instances to a capital "P". This should be corrected back to "C. psittaci".

Response: Thanks for your kind remind. It has been changed to “*C. psittaci*” throughout the manuscript.

2. Lines 247-256 - change "colonized" to "colonizing"

Response: Sorry for the trouble caused by our inaccurate expression. Thanks very much for your kind remind. We have changed it to "*colonizing*".

Special thanks to you for your professional and kind advice and comments. We hope the responses to reviewer's comments and revised manuscript could provide relatively reasonable and satisfactory replies to the comments of reviewer and reach the demand of publication.

Thanks again for your work and giving us opportunity for revising again!

Best regards!

Sincerely,

Na Li

May 24, 2022

Prof. Na Li
Second Affiliated Hospital of Chongqing Medical University
chongqing
China

Re: Spectrum02384-21R2 (The application of metagenomic next-generation sequencing in the diagnosis of pneumonia caused by *Chlamydia psittaci*)

Dear Prof. Na Li:

Link Not Available

Sincerely,

Hui Wang

Journals Department
Reviewer comments:

Reviewer #1 (Comments for the Author):

Thank you for thoroughly addressing all reviewer comments and making the requested changes. My only remaining minor comment is to move the new text in lines 506-517 to the results, since SDSMRN criteria and cutoffs are not mentioned previously in the manuscript. Also, please define the acronym SDSMRN.

Staff Comments:

Preparing Revision Guidelines

Please return the manuscript within 60 days; if you cannot complete the modification within this time period, please contact me. If you do not wish to modify the manuscript and prefer to submit it to another journal, please notify me of your decision immediately so that the manuscript may be formally withdrawn from consideration by Microbiology Spectrum.

Response to Reviewers

Re: Manuscript ID: Spectrum02384-21R2, and Title: "The application of metagenomic next-generation sequencing in the diagnosis of pneumonia caused by *Chlamydia psittaci*."

Dear editor and reviewers,

Thanks for your letter and for the reviewers' comments concerning our manuscript entitled "**The application of metagenomic next-generation sequencing in the diagnosis of pneumonia caused by *Chlamydia psittaci*.**" (Manuscript ID: **Spectrum02384-21R2**). Thank you very much for your re-reading of our revised article. We are very grateful for your professional and responsible attitude and your hard work on our article. We appreciate a lot for the work of the reviewer and the precious comments to our manuscript. We have studied the comments carefully and have made corrections. Revised portions are marked in red in the paper. The main corrections in the paper and the responses to the reviewer's comments are as follows:

Responses to the reviewer's comments:

Reviewer #1:

Thank you for thoroughly addressing all reviewer comments and making the requested changes.

My only remaining minor comment is to move the new text in lines 506-517 to the results, since SDSMRN criteria and cutoffs are not mentioned previously in the manuscript. Also, please define the acronym SDSMRN.

Response: Thanks very much for your recognition of our revised article. Thank you for your professional work and kind suggestions on the article. We have moved the new text in Lines 506-517 to the results in Lines 261-173 in the revised manuscript and defined SDSMRN as **standardized specifically mapped read number** in Line 268 in the revised manuscript. Thank you again for your patience and careful work.

Special thanks again to you for your professional and kind advices and comments. We are very happy to communicate with you, so that we get more valuable experience in improving our manuscript and learn a lot of professional knowledge. We hope the response to reviewer's comments and revised manuscript could provide relatively reasonable and satisfactory replies to the comments of reviewer and reach the demand of publication.

Thanks again for your work and giving us opportunity for revising again!

Best regards!

Sincerely,

Na Li

June 6, 2022

Prof. Na Li
Second Affiliated Hospital of Chongqing Medical University
chongqing
China

Re: Spectrum02384-21R3 (The application of metagenomic next-generation sequencing in the diagnosis of pneumonia caused by *Chlamydia psittaci*)

Dear Prof. Na Li:

Your manuscript has been accepted, and I am forwarding it to the ASM Journals Department for publication. You will be notified when your proofs are ready to be viewed.

Sincerely,

Hui Wang
Editor, Microbiology Spectrum

Journals Department
Appedix1: Accept